# On the Approximation Errors of Node Sampling for Graph Neural Networks

## Abstract

The recent advancements in graph neural networks (GNNs) have led to state-of-the-art performances in various applications, including chemo-informatics, question-answering systems, and recommender systems. However, scaling up these methods to huge graphs such as social network graphs and web graphs still remains a challenge. In particular, the existing methods for accelerating GNNs are either not theoretically guaranteed in terms of approximation error, or they require at least a linear time computation cost. In this study, we reveal the query complexity of the uniform node sampling scheme for GraphSAGE, the graph attention networks (GAT), and the graph convolutional networks (GCN). The key advantage of our analysis is that the complexity is completely independent of the numbers of the nodes, edges, and neighbors of the input and depends only on the error tolerance and confidence probability while providing theoretical guarantee for the approximation error. To the best of our knowledge, this is the first work to give a theoretical guarantee of approximation for GNNs within a constant time complexity. Through experiments using synthetic and real-world datasets, we demonstrate the speed and precision of the node sampling scheme and validate our theoretical results.

## 1 Introduction

Machine learning on graph structures has various applications such as chemo-informatics (Gilmer et al., 2017), question answering systems (Schlichtkrull et al., 2018), and recommender systems (Fan et al., 2019). Recently, a novel machine learning model for graph data called graph neural networks (GNNs) (Gori et al., 2005; Scarselli et al., 2009; Kipf & Welling, 2017; Hamilton et al., 2017) demonstrated state-of-the-art performances in various graph learning tasks. However, large scale graphs such as social network graphs and web graphs contain billions of nodes, and even a linear time computation cost per iteration is prohibited. Therefore, applying GNNs to huge graphs is challenging. Although Ying et al. (2018) succeeded in applying GNNs to a web-scale network using MapReduce, it still requires massive computational resources.

There are several node sampling techniques to reduce GNN computation. For example, an empirical neighbor sampling scheme is used to speed up GraphSAGE (Hamilton et al., 2017). FastGCN employs a random layer-wise node sampling (Chen et al., 2018b). Huang et al. (2018) further improved FastGCN by using an adaptive sampling technique to reduce the variance of estimators. Chen et al. (2018a) proposed a variant of neighbor sampling, which used historical activations to reduce the estimator variance. Overall, the existing sampling techniques for GNNs work well in practice. However, these techniques are either not theoretically guaranteed in terms of approximation error, or thy require at least a linear time computation cost.

In this study, we consider the problem of approximating the embedding of *one* node using GNNs *in constant time* with maximum precision[1]. We analyze the neighbor sampling technique (Hamilton et al., 2017) to show that a constant number of samples are needed to guarantee the approximation error. It should be noted that the neighbor sampling was introduced as a heuristic method originally, and they did not provide any theoretical guarantees. Specifically, given an error tolerance $\varepsilon$ and

---

[1]e.g., predicting whether a user of an SNS clicks an advertisement by GNNs in real-time (i.e., when the user accesses). A user may have many neighbors, but GNNs must respond in limited time, which prohibits exact computation. It motivates us to approximate the exact computation in limited time with theoretical guarantee.

Table 1: ✓ indicates *neighbor sampling* approximates the network in constant time. ✗ indicates *any algorithm* cannot approximate the network in constant time. ✓ in the Gradient column indicates the error between the gradient of the approximated embedding and that of the exact embedding is also theoretically bounded. ✓* needs an additional condition to approximate it in constant time.

| Activation | GAT, GraphSAGE-{GCN, mean} | | GraphSAGE-pool | GCN | |
| --- | --- | --- | --- | --- | --- |
| | Embedding | Gradient | | Embedding | Gradient |
| sigmoid / tanh | ✓ Thm. 1 | ✓ Thm. 4 | ✗ Thm. 9 | ✓* Thm. 1 | ✓* Thm. 4 |
| ReLU | ✓ Thm. 1 | ✗ Thm. 8 | ✗ Thm. 9 | ✓* Thm. 1 | ✗ Thm. 8 |
| ReLU + normalization | ✗ Thm. 7 | ✗ Thm. 7 | ✗ Thm. 9 | ✗ Thm. 7 | ✗ Thm. 7 |

confidence probability $1 - \delta$, our analysis shows that the estimate $\hat{z}_v$ of the exact embedding $z_v$ of a node $v$ such that $\Pr[\|\hat{z}_v - z_v\|_2 \geq \varepsilon] \leq \delta$ and the estimate $\widehat{\frac{\partial z_v}{\partial \theta}}$ of the exact gradient $\frac{\partial z_v}{\partial \theta}$ of the embedding $z_v$ with respect to the network parameters $\theta$, such that $\Pr[\|\widehat{\frac{\partial z_v}{\partial \theta}} - \frac{\partial z_v}{\partial \theta}\|_F \geq \varepsilon] \leq \delta$ can be computed in a constant time. Especially, the uniform node sampling can approximate the exact embedding and its gradients within $O(\frac{1}{\varepsilon^{2L}} (\log \frac{1}{\varepsilon} + \log \frac{1}{\delta})^{L-1} \log \frac{1}{\delta})$ time, where $L$ denotes the number of layers. This complexity is completely independent of the number of nodes, edges, and neighbors of the input, which enables us to deal with graphs irrespective of their size. Moreover, the complexity is a polynomial with respect to $\frac{1}{\varepsilon}$ and $\log \frac{1}{\delta}$. We demonstrate that the time complexity is optimal when $L = 1$ with respect to the error tolerance $\varepsilon$.

Through experiments, we show that the approximation error between the exact computation and its approximation rapidly converges to zero. To the best of our knowledge, this is the first constant time approximation algorithm for GNNs with a theoretical guarantee in terms of approximation error.

**Contributions:** The contributions of this paper are summarized as follows:

- We analyze the neighbor sampling technique for GraphSAGE, GAT, and GCN to provide theoretical justification. Especially, our analysis shows that the complexity is completely independent of the number of nodes, edges, and neighbors of the input.
- We show that some existing GNNs, including the original GraphSAGE (Hamilton et al., 2017), cannot be approximated in constant time by any algorithm (see Table 1 for details).
- We empirically validate our theorems using synthetic and real-world datasets.

## 2 RELATED WORK

### 2.1 GRAPH NEURAL NETWORKS

Graph neural networks (GNNs) were first introduced by Gori et al. (2005) and Scarselli et al. (2009). They obtained node embedding by recursively applying the propagation function until convergence. Kipf & Welling (2017) proposed graph convolutional networks (GCN), which significantly outperformed the existing methods, including non-neural network based approaches. Gilmer et al. (2017) proposed the message passing neural networks (MPNNs), a general framework of GNNs using the message passing mechanism. Veličković et al. (2018) proposed the graph attention networks (GAT), which incorporate the attention mechanism into GNNs. With the advent of GAT, various GNN models with the attention mechanism have been recently proposed (Wang et al., 2019; Park et al., 2019).

GraphSAGE (Hamilton et al., 2017) is another GNN model, which employs neighbor sampling to reduce the computational costs of training and inference. Owing to neighbor sampling, GraphSAGE can deal with large graphs. However, neighbor sampling was introduced without any theoretical guarantee, and the number of samples is chosen empirically. An alternative computationally efficient GNN would be FastGCN (Chen et al., 2018b), which employs layer-wise random node sampling to speed up training and inference. Huang et al. (2018) further improved FastGCN by using an adaptive node sampling technique to reduce the variance of estimators. Thanks to the adaptive

sampling technique, it reduces the computational costs and outperforms neighbor sampling in terms of classification accuracy and convergence speed. Chen et al. (2018a) proposed an alternative neighbor sampling technique, which uses historical activations to reduce the estimator variance. Additionally, it could achieve zero variance after a certain number of iterations. However, because it used the same sampling technique of GraphSAGE to obtain the initial solution, the approximation error was not theoretically bounded until the $\Omega(n)$-th iteration. Overall, the existing sampling techniques work well in practice. However, these techniques are either not theoretically guaranteed in terms of approximation error, or they require at least a linear time computation cost to calculate the embedding of a node and its gradient of GNN models. Moreover, it is not clear whether we can apply the sampling techniques to state-of-the-art GNN models such as GAT.

## 2.2 SUBLINEAR TIME ALGORITHMS

The sublinear time algorithms were originally proposed for property testing (Rubinfeld & Sudan, 1996). Sublinear property testing algorithms check whether the input has some property $\pi$ or the input is sufficiently far from the property $\pi$ with high probability in sublinear time with respect to the input size. Sublinear time approximation algorithms are another type of sublinear time algorithms. More specifically, they calculate a value sufficiently close to the exact value with high probability in sublinear time. Constant time algorithms are a subclass of sublinear time algorithms. They work not only in sublinear time with respect to the input size but also in constant time. The proposed algorithm is classified as a constant time approximation algorithm.

The examples of sublinear time approximation algorithms include minimum spanning tree in metric space (Czumaj & Sohler, 2004) and minimum spanning tree with integer weights (Chazelle et al., 2005). Parnas & Ron (2007) proposed a method to convert distributed local algorithms into constant time approximation algorithms. In their study, they proposed a method to construct constant time algorithms for the minimum vertex cover problem and dominating set problem. A classic example of sublinear time algorithms related to machine learning includes clustering (Indyk, 1999; Mishra et al., 2001). Examples of recent work in this stream include constant time approximation of the minimum value of quadratic functions (Hayashi & Yoshida, 2016) and constant time approximation of the residual error of the Tucker decomposition (Hayashi & Yoshida, 2017). They adopted simple sampling strategies to obtain theoretical guarantee similar to our work.

In this paper, we provide theoretical guarantee for approximation of GNNs within a constant time for the first time.

## 3 BACKGROUND

### 3.1 NOTATIONS

Let $G$ be the input graph, $\mathcal{V} = \{1, 2, \ldots, n\}$ be the set of nodes, $n = |\mathcal{V}|$ be the number of nodes, $\mathcal{E}$ be the set of edges, $m = |\mathcal{E}|$ be the number of edges, $\deg(v)$ be the degree of a node $v$, $\mathcal{N}(v)$ be the set of neighbors of a node $v$, $\boldsymbol{x}_v \in \mathbb{R}^{d_0}$ be the feature vector associated to a node $v \in \mathcal{V}$, $\boldsymbol{X} = (\boldsymbol{x}_1, \boldsymbol{x}_2 \ldots, \boldsymbol{x}_n)^\top \in \mathbb{R}^{n \times d_0}$ be the stacked feature vectors, and $^\top$ denotes the matrix transpose.

### 3.2 NODE EMBEDDING MODEL

We consider the node embedding problem using GNNs. Especially, we employ the message passing neural networks (MPNNs) framework (Gilmer et al., 2017). This framework includes many GNN models, such as GraphSAGE and GCN. Algorithm 1 shows the algorithm of MPNNs. We refer to $\boldsymbol{z}_v^{(L)}$ as $\boldsymbol{z}_v$ simply. The aim of this study is to develop a constant time approximation algorithm for calculating the embedding vector $\boldsymbol{z}_v$ and gradients $\frac{\partial \boldsymbol{z}_v}{\partial \boldsymbol{\theta}}$ with the given model parameters $\boldsymbol{\theta}$ and node $v$.

---

**Algorithm 1** $\mathcal{O}_z$: Exact embedding

**Require:** Graph $G = (\mathcal{V}, \mathcal{E})$; Features $\boldsymbol{X} \in \mathbb{R}^{n \times d_0}$; Node index $v \in \mathcal{V}$; Model parameters $\boldsymbol{\theta}$.
**Ensure:** Exact embedding $\boldsymbol{z}_v$

1: $\boldsymbol{z}_i^{(0)} \leftarrow \boldsymbol{x}_i \ (\forall i \in \mathcal{V})$
2: **for** $l \in \{1, \ldots, L\}$ **do**
3:   **for** $i \in \mathcal{V}$ **do**
4:     $\boldsymbol{h}_i^{(l)} \leftarrow \sum_{u \in \mathcal{N}(i)} M_{liu}(\boldsymbol{z}_i^{(l-1)}, \boldsymbol{z}_u^{(l-1)}, \boldsymbol{e}_{iu}, \boldsymbol{\theta})$
5:     $\boldsymbol{z}_i^{(l)} \leftarrow U_l(\boldsymbol{z}_i^{(l-1)}, \boldsymbol{h}_i^{(l)}, \boldsymbol{\theta})$
6:   **end for**
7: **end for**
8: **return** $\boldsymbol{z}_v^{(L)}$

---

### 3.3 COMPUTATIONAL MODEL ASSUMPTIONS

We have to specify how to access the input to design constant time algorithms because the constant time algorithms cannot read the entire input. We follow the standard convention of sublinear time algorithms (Parnas & Ron, 2007; Nguyen & Onak, 2008). We model our algorithm as an oracle machine that can generate queries regarding the input, and we measure the complexity by query complexity. Algorithms can access the input only by querying the following oracles: (1) $\mathcal{O}_{\text{deg}}(v)$: the degree of node $v$, (2) $\mathcal{O}_G(v, i)$: the $i$-th neighbor of node $v$, and (3) $\mathcal{O}_{\text{feature}}(v)$: the feature of node $v$. We assume that our algorithm can query the oracles in constant time per query.

### 3.4 PROBLEM FORMULATION

Given a node $v$, we calculate the following functions with the least number of oracle accesses: (1) $\mathcal{O}_{\boldsymbol{z}}(v)$: the embedding $\boldsymbol{z}_v$ and (2) $\mathcal{O}_g(v)$: the gradients of parameters $\frac{\partial \boldsymbol{z}_v}{\partial \boldsymbol{\theta}}$. However, the exact computation of $\mathcal{O}_z$ and $\mathcal{O}_g$ needs at least $\deg(v)$ queries to aggregate the features from the neighbor nodes, which is not constant time with respect to the input size. Thus, it is computationally expensive to execute the algorithm for a huge network. Therefore, we consider making the following approximations:

- $\hat{\mathcal{O}}_z(v, \varepsilon, \delta)$: an estimate $\hat{\boldsymbol{z}}_v$ of $\boldsymbol{z}_v$ such that $\Pr[\|\hat{\boldsymbol{z}}_i - \boldsymbol{z}_i\|_2 \geq \varepsilon] \leq \delta$,

- $\hat{\mathcal{O}}_g(v, \varepsilon, \delta)$: an estimate $\widehat{\frac{\partial \boldsymbol{z}_v}{\partial \boldsymbol{\theta}}}$ of $\frac{\partial \boldsymbol{z}_v}{\partial \boldsymbol{\theta}}$ such that $\Pr[\|\widehat{\frac{\partial \boldsymbol{z}_v}{\partial \boldsymbol{\theta}}} - \frac{\partial \boldsymbol{z}_v}{\partial \boldsymbol{\theta}}\|_F \geq \varepsilon] \leq \delta$,

where $\varepsilon > 0$ is the error tolerance, $1 - \delta$ is the confidence probability, and $\| \cdot \|_2$ and $\| \cdot \|_F$ are the Euclidean and Frobenius norm, respectively.

Under the fixed model structure (i.e., the number of layers $L$, the message passing functions, and the update functions), we construct an algorithm that calculates $\hat{\mathcal{O}}_z$ and $\hat{\mathcal{O}}_g$ in constant time irrespective of the number of nodes, edges, and neighbors of the input.

However, it is impossible to construct a constant time algorithm without any assumption about the inputs. Therefore, we make the following mild assumptions:

**Assumption 1** $\exists B \in \mathbb{R}$ s.t. $\|\boldsymbol{x}_i\|_2 \leq B$, $\|\boldsymbol{e}_{iu}\|_2 \leq B$, and $\|\boldsymbol{\theta}\|_2 \leq B$.

**Assumption 2** $\deg(i)M_{liu}$ and $U_l$ are uniformly continuous in any bounded domain.

**Assumption 3** (Only for gradient computation) $\deg(i)DM_{liu}$ and $DU_l$ are uniformly continuous in any bounded domain, where $D$ denotes the Jacobian operator.

Building a constant time algorithm is impossible without these assumptions as shown in Section 5. Furthermore, we derive the query complexity of our algorithm when the message functions and update functions satisfy the following assumptions.

**Assumption 4** $\exists K \in \mathbb{R}$ s.t. $\deg(i)M_{liu}$ and $U_l$ are $K$-Lipschitz continuous in any bounded domain.

**Assumption 5** (Only for gradient computation) $\exists K' \in \mathbb{R}$ s.t. $\deg(i)DM_{liu}$ and $DU_l$ are $K'$-Lipschitz continuous in any bounded domain.

## 4 PROPOSED METHOD

### 4.1 CONSTANT TIME EMBEDDING APPROXIMATION

Here, we build a constant time approximation algorithm based on neighbor sampling, which approximates the embedding $\boldsymbol{z}_v$ with an absolute error of at most $\varepsilon$ and probability $1 - \delta$. We recursively construct the algorithm layer by layer by sampling $r^{(l)}$ neighboring nodes in layer $l$. We refer to the algorithm that calculates the estimate of embeddings in the $l$-th layer $\boldsymbol{z}^{(l)}$ as $\hat{\mathcal{O}}_z^{(l)}(l = 1, \ldots, L)$. The pseudo code is presented in Algorithm 2. Here, $\hat{\mathcal{O}}_z^{(l-1)}(v \leftarrow u)$ means calling the function $\hat{\mathcal{O}}_z^{(l-1)}$ with the same parameters as the current function except for $v$, which is replaced by $u$. In the following, we demonstrate the theoretical properties of Algorithm 2.

---

**Algorithm 2** $\hat{\mathcal{O}}_z^{(l)}$: Estimate the embedding $\boldsymbol{z}_v^{(l)}$

---

**Require:** Graph $G = (\mathcal{V}, \mathcal{E})$ (as oracle); Features $\boldsymbol{X} \in \mathbb{R}^{n \times d_0}$ (as oracle); Node index $v \in \mathcal{V}$;
    Model parameters $\boldsymbol{\theta}$; Error tolerance $\varepsilon$; Confidence probability $1 - \delta$.
**Ensure:** Approximation of the embedding $\boldsymbol{z}_v^{(l)}$
  1: $\mathcal{S}^{(l)} \leftarrow$ sample $r^{(l)}(\varepsilon, \delta)$ neighbors of $v$ with uniform random with replacement.
  2: $\hat{\boldsymbol{h}}_v^{(l)} \leftarrow \begin{cases} \frac{\mathcal{O}_{\text{deg}}(v)}{r^{(l)}} \sum_{u \in \mathcal{S}^{(l)}} M_{lvu}(\mathcal{O}_{\text{feature}}(v), \mathcal{O}_{\text{feature}}(u), \boldsymbol{e}_{iu}, \boldsymbol{\theta}) & (l = 1) \\ \frac{\mathcal{O}_{\text{deg}}(v)}{r^{(l)}} \sum_{u \in \mathcal{S}^{(l)}} M_{lvu}(\hat{\mathcal{O}}_z^{(l-1)}(v \leftarrow v), \hat{\mathcal{O}}_z^{(l-1)}(v \leftarrow u), \boldsymbol{e}_{iu}, \boldsymbol{\theta}) & (l > 1) \end{cases}$
  3: $\hat{\boldsymbol{z}}_v^{(l)} \leftarrow U_l(\hat{\mathcal{O}}_z^{(l-1)}(v \leftarrow v), \hat{\boldsymbol{h}}_v^{(l)}, \boldsymbol{\theta})$ if $l > 1$ otherwise $U_l(\mathcal{O}_{\text{feature}}(v), \hat{\boldsymbol{h}}_v^{(l)}, \boldsymbol{\theta})$
  4: **return** $\hat{\boldsymbol{z}}_i$

---

**Theorem 1.** *For all $\varepsilon > 0, 1 > \delta > 0$, there exists $r^{(l)}(\varepsilon, \delta)$ $(l = 1, \ldots, L)$ such that for all inputs satisfying Assumptions 1 and 2, the following property holds true:*

$$\Pr[\|\hat{\mathcal{O}}_z(v, \varepsilon, \delta) - \boldsymbol{z}_v\|_2 \geq \varepsilon] \leq \delta.$$

Theorem 1 shows that the approximation error of Algorithms 2 is bounded by $\varepsilon$ with the probability $1 - \delta$. It is proved by Hoeffding's inequality (Hoeffding, 1963). The complete proof is available in the appendices. Because the number of sampled nodes depends only on $\varepsilon$ and $\delta$ and independent of the number of nodes, edges, and neighbors of the input, Algorithm 2 works in constant time. We provide the complexity when the functions are Lipschitz continuous.

**Theorem 2.** *Under Assumption 1 and 4, $r^{(L)} = O(\frac{1}{\varepsilon^2} \log \frac{1}{\delta})$ and $r^{(1)}, \ldots, r^{(L-1)} = O(\frac{1}{\varepsilon^2}(\log \frac{1}{\varepsilon} + \log \frac{1}{\delta}))$ are sufficient, and the query complexity of Algorithms 2 is $O(\frac{1}{\varepsilon^{2L}}(\log \frac{1}{\varepsilon} + \log \frac{1}{\delta})^{L-1} \log \frac{1}{\delta})$.*

We show that the query complexity of Algorithm 2 is optimal with respect to $\varepsilon$ if the number of layers is one. In other words, a one-layer model cannot be approximated in $o(\frac{1}{\varepsilon^2})$ time by any algorithm.

**Theorem 3.** *Under Assumptions 1 and 4 and $L = 1$, the time complexity of Algorithm 2 in Theorem 2 is optimal with respect to the error tolerance $\varepsilon$.*

The proof is based on Chazelle et al.'s lemma (Chazelle et al., 2005). The optimality when $L \geq 2$ is an open problem.

### 4.2 Constant Time Gradient Approximation

Next, we propose a constant time algorithm that approximates the gradient of embeddings with respect to the model parameters with an absolute error of at most $\varepsilon$ and probability $1 - \delta$. The basic strategy is to execute Algorithm 2 and calculate the gradients of the embedding $\boldsymbol{z}_v$. Let $\frac{\partial \boldsymbol{z}_v}{\partial \boldsymbol{\theta}}$ be the gradient of the embedding $\boldsymbol{z}_v$ with respect to the model parameter $\boldsymbol{\theta}$ (i.e., $(\frac{\partial \boldsymbol{z}_v}{\partial \boldsymbol{\theta}})_{ijk} = \frac{\partial \boldsymbol{z}_{vi}}{\partial \boldsymbol{\theta}_{jk}}$).

**Theorem 4.** *For all $\varepsilon > 0, 1 > \delta > 0$, there exists $r^{(l)}(\varepsilon, \delta)$ $(l = 1, \ldots, L)$ such that for all inputs satisfying Assumptions 1, 2, and 3, the following property holds true:*

$$\Pr[\|\widehat{\frac{\partial \boldsymbol{z}_v^{(L)}}{\partial \boldsymbol{\theta}}} - \frac{\partial \boldsymbol{z}_v^{(L)}}{\partial \boldsymbol{\theta}}\|_F \geq \varepsilon] \leq \delta,$$

*where $\widehat{\frac{\partial \boldsymbol{z}_v^{(L)}}{\partial \boldsymbol{\theta}}}$ is the gradient of $\hat{\boldsymbol{z}}_v^{(L)}$, which is obtained by $\hat{\mathcal{O}}_z^{(L)}(v, \varepsilon, \delta)$, with respect to $\boldsymbol{\theta}$.*

Therefore, we can calculate an estimate of the gradient of the embedding with respect to parameters with an absolute error of at most $\varepsilon$ and probability $1 - \delta$ by running $\hat{\mathcal{O}}_z^{(L)}(v, \varepsilon, \delta)$ and calculating the gradient of the obtained estimate of the embedding. We provide the complexity when the functions are Lipschitz continuous.

**Theorem 5.** *Under Assumptions 1, 4, and 5, $r^{(L)} = O(\frac{1}{\varepsilon^2} \log \frac{1}{\delta})$ and $r^{(1)}, \ldots, r^{(L-1)} = O(\frac{1}{\varepsilon^2}(\log \frac{1}{\varepsilon} + \log \frac{1}{\delta}))$ are sufficient, and the gradient of the embedding with respect to parameters can be approximated with an absolute error of at most $\varepsilon$ and probability $1 - \delta$ in $O(\frac{1}{\varepsilon^{2L}}(\log \frac{1}{\varepsilon} + \log \frac{1}{\delta})^{L-1} \log \frac{1}{\delta})$ time.*

Note that, technically, MPNNs do not include GAT because GAT uses embeddings of other neighboring nodes to calculate the attention value. However, Theorem 1 and 4 can be naturally extended to GAT, and we can approximate GAT in constant time by neighbor sampling. The details are described in the appendices.

## 5 INAPPROXIMABILITY

In this section, we show that some existing GNNs cannot be approximated in constant time. These theorems state that these models cannot be approximated in constant time not only by our algorithm but also *by any other algorithm*. In other words, for any algorithm that works in constant time, there exists an error tolerance $\varepsilon$, a confidence probability $1 - \delta$, and a counter example input such that the approximation error for the input is more than $\varepsilon$ with probability $\delta$. It indicates that the application of an approximation method to these models requires great supervision because the obtained embedding may be significantly different from the exact embedding.

**Theorem 6.** *If $\|\boldsymbol{x}_i\|_2$ or $\|\boldsymbol{\theta}\|_F$ is not bounded, even under Assumption 1 and 2, the embeddings of GraphSAGE-GCN cannot be approximated with arbitrary precision and probability in constant time.*

**Theorem 7.** *Even under Assumption 1, the embeddings and gradients of GraphSAGE-GCN with ReLU activation and normalization cannot be approximated with arbitrary precision and probability in constant time.*

We confirm Theorem 7 through computational experiments in Section 6.

**Theorem 8.** *Even under Assumptions 1 and 2, the gradients of GraphSAGE-GCN with ReLU activation cannot be approximated with arbitrary precision and probability in constant time.*

However, it should be noted that the *embeddings* of GraphSAGE-GCN with ReLU activation (without normalization layer) *can* be approximated in constant time using our algorithm by Theorem 1. The following two theorems state that these models cannot be approximated in constant time even under Assumptions 1, 2, and 3.

**Theorem 9.** *Even under assumptions 1, 2, and 3, the embeddings of GraphSAGE-pool cannot be approximated with arbitrary precision and probability in constant time.*

**Theorem 10.** *Even under Assumptions 1, 2, and 3, the embeddings of GCN cannot be approximated with arbitrary precision and probability in constant time.*

**Constant Time Approximaiton for GCN:** Due to the inapproximability theorems, we cannot approximate GCN in constant time. However, we can approximate GCN in constant time if the input graph satisfies the following property:

**Assumption 6** There exists a constant $C \in \mathbb{R}$ such that for any input graph $G = (\mathcal{V}, \mathcal{E})$ and node $v, u \in \mathcal{V}$, the ratio of $\deg(v)$ to $\deg(u)$ is at most $C$ (i.e., $\deg(v)/\deg(u) \le C$).

Assumption 6 prohibits input graphs that have a skewed degree distribution. GCN needs Assumption 6 because (1) the norm of the embedding is not bounded and (2) the influence of anomaly nodes with low degrees is significant without Assumption 6. It should be noted that the GraphSAGE-pool cannot be approximated in constant time even under Assumption 6.

## 6 EXPERIMENTS

We will answer the following questions through experiments:

**Q1:** How fast is the constant time approximation algorithm (Algorithm 2)?

**Q2:** Does Algorithm 2 accurately approximate the embeddings of GraphSAGE-GCN without normalization (Theorem 1), whereas it cannot approximate the original one (Theorem 7)?

**Q3:** Does Algorithm 2 accurately approximate the gradients of GraphSAGE-GCN with sigmoid (Theorem 4), whereas it cannot approximate that with ReLU activation (Theorem 8)?

**Q4:** Is the theoretical rate of the approximation error of Algorithm 2 tight?

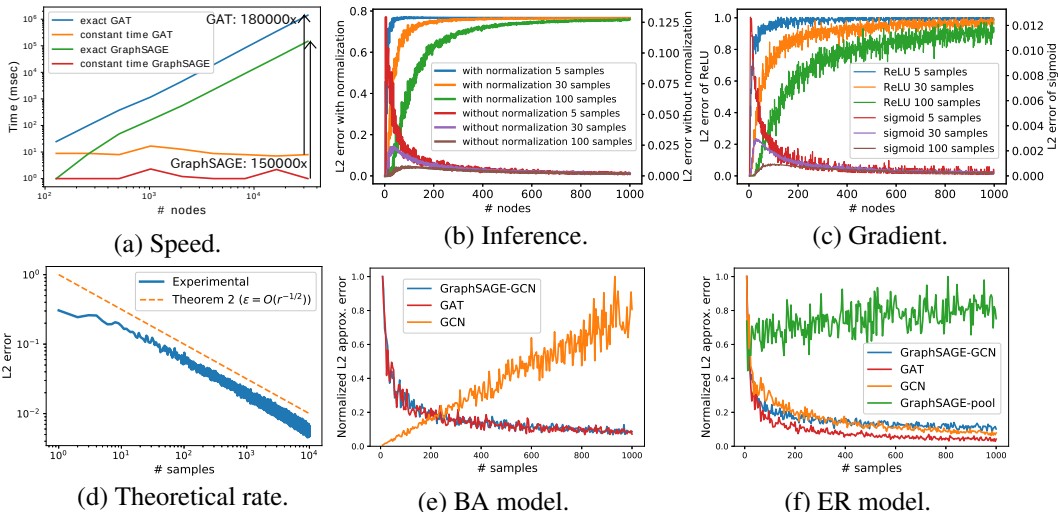

Figure 1: (a) The inference speed of each method. (b) The approximation error of the original GraphSAGE-GCN (i.e., with ReLU activation and normalization) and GraphSAGE-GCN with ReLU activation. (c) The approximation error of the gradient with ReLU and sigmoid activations. (d) The approximation error of Algorithm 2 and its theoretical bound. (e) The approximation error of GraphSAGE-GCN, GAT, and GCN with the Barabasi-Albert model. (f) The approximation error of GraphSAGE-GCN, GAT, GCN, and GraphSAGE-pool with the Erdős-Rényi model.

**Q5:** Does neighbor sampling fail to approximate GCN when the degree distribution is skewed (Theorem 10)? Does it succeed when the node distribution is flat (Assumption 6)?

**Q6:** Does neighbor sampling efficiently work for real data?

It should be noted that we focus on showing the approximation error of neighbor sampling in this study as Hamilton et al. (2017) have already reported the effect of neighbor sampling for downstream machine learning tasks (e.g., classification). The experimental details are described in the appendices.

**Experiments for Q1:** We measure the speed of exact computation and the constant time approximation of the two-layer GraphSAGE-GCN and two-layer GAT. We initialize parameters using the i.i.d. standard multivariate normal distribution. The input graph is a clique $K_n$. We use ten-dimensional vectors from the i.i.d. standard multivariate normal distribution as the node features. We take $r^{(1)} = r^{(2)} = 100$ samples. Figure 1 (a) shows the speed of these methods as the number of nodes increases. This shows that the constant time approximation is several orders of magnitude faster than the exact computation when the number of nodes is large.

**Experiments for Q2:** We use the original one-layer GraphSAGE-GCN (with ReLU activation and normalization) and one-layer GraphSAGE-GCN with ReLU activation. The input graph is a clique $K_n$, whose features are $\boldsymbol{x}_1 = (1,0)^\top$ and $\boldsymbol{x}_i = (0,1/n)^\top$ $(i \neq 1)$, and the weight matrix is an identity matrix $\boldsymbol{I}_2$. We use $r^{(1)} = 5, 30$, and $100$ as the sample size. If a model can be approximated in constant time, the approximation error goes to zero as the sample size increases even if the graph size reaches infinity. The approximation errors of both models are illustrated in Figure 1 (b). The approximation error of the original GraphSAGE-GCN converges to approximately $0.75$ even if the sample size increases. In contrast, the approximation error without normalization becomes increasingly bounded as the sample size increases. This is consistent with Theorems 1 and 7.

**Experiments for Q3:** We study the approximation errors of the gradients. We use the one-layer GraphSAGE-GCN with ReLU activation and sigmoid activation. The input graph is a clique $K_n$, whose features are $\boldsymbol{x}_1 = (1,2)^\top$ and $\boldsymbol{x}_i = (1,1)^\top$ $(i \neq 1)$, and the weight matrix is $((-1,1))$. We use $r^{(1)} = 5, 30$, and $100$ as the sample size. The approximation error of both models are illustrated in Figure 1 (c). The approximation error with ReLU activation converges to approximately $1.0$ even if the sample size increases. In contrast, the approximation error with sigmoid activation becomes increasingly bounded as the sample size increases. This is consistent with Theorems 4 and 8.

**Experiments for Q4:** We use the one-layer GraphSAGE-GCN with sigmoid activation. The input graph is a clique $K_n$, where the number of nodes is $n = 40000$. We set the dimensions of intermediate embeddings as 2, and each feature value is set to 1 with probability 0.5 and $-1$ otherwise. This satisfies Assumption 1 (i.e., $\|x_i\|_2 \leq \sqrt{2}$). We compute the approximation errors of Algorithm 2 with different numbers of samples. Figure 1 (d) illustrates the 99-th percentile point of empirical approximation errors and the theoretical bound by Theorem 2 (i.e., $\varepsilon = O(r^{-1/2})$). It shows that the approximation error decreases along with the theoretical rate. It indicates the theoretical rate is tight.

**Experiments for Q5:** We analyze the instances when neighbor sampling succeeds and fails for a variety of models. First, we use the Barabasi-Albert model (BA model) (Barabasi & Albert, 1999) to generate input graphs. The degree distribution of the BA model follows a power-law. Therefore, neighbor sampling will fail to approximate GCN (Theorem 10, Assumption 6). We use two-layer GraphSAGE-GCN, GAT, and GCN with ReLU activation. We use ten-dimensional vectors from the i.i.d. standard multivariate normal distribution as the node features. In this experiment, we use the same number $r$ of samples in the first and second layer (i.e., $r = r^{(1)} = r^{(2)}$) and we change $r$ from 8 to 1000. We use graphs with $n = r^2$ nodes. Note that if the constant time approximation is possible, the error is bounded as the number of samples increases even if the number of nodes increases. Figure 1 (e) shows the approximation error. It shows that the error of GCN linearly increases even if the number of samples increases. However, the errors of GraphSAGE-GCN and GAT gradually decrease as the number of samples increases. It indicates that we cannot bound the approximation error of GCN however large number of examples we use. This is consistent with Theorem 10. This result indicates that approximating GCN requires great supervision when the input graph is a social network because the degree distribution of a social network presents the power-law as the BA-model.

Next, we use the Erdős-Rényi model (ER model) (Erdős & Rényi, 1959). It generates graphs with flat degree distribution. We use the two-layer GraphSAGE-GCN, GAT, GCN, and GraphSAGE-pool. Figure 1 (f) shows the approximation error. It shows that the errors of GraphSAGE-GCN, GAT, and GCN gradually decrease as the number of samples increases. This is consistent with Theorem 1 and Assumption 6. In contrast, the approximation error of GraphSAGE-pool does not decrease even if the input graphs are generated by the ER model. This is consistent with Theorem 9.

**Experiments for Q6:** We use three real-world datasets: Cora, PubMed, and Reddit. They contain 2708, 19717, and 232965 nodes, respectively. We randomly choose 500 nodes for validation and 1000 nodes for test and use the remaining nodes for training. We use the two-layer GraphSAGE-GCN with sigmoid activation in this experiment. The dimensions of the hidden layers are set to 128, and we use an additional fully connected layer to predict the labels of the nodes from the embeddings. We train the models with Adam (Kingma & Ba, 2015) with a learning rate of 0.001. We first train ten models with training nodes for each dataset. The micro-F1 scores of Cora, PubMed, and Reddit are 0.877, 0.839,

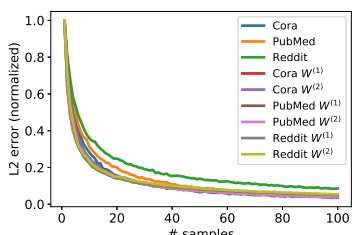

Figure 2: Real data.

and 0.901, respectively. It should be noted that we do not aim to obtain high classification accuracy here but intend to sanity check the models. In this experiment, we use the same number $r$ of samples in the first and second layer (i.e., $r = r^{(1)} = r^{(2)}$) and we change $r$ from 1 to 100. We also calculate the gradient of parameter matrices $W^{(1)}$ and $W^{(2)}$ with respect to the embedding obtained by Algorithm 2. We compare these values with exact embeddings and exact gradients. Figure 6 illustrates the 99-th percentile point of the empirical approximation errors. We normalize them to ensure that the value is 1.0 at $r = 1$ to demonstrate the decreasing rate of each error. It shows that the approximation errors of embeddings and gradients rapidly decrease for the real-world data.

## 7 CONCLUSION

We proposed a constant time approximation algorithm for the embedding and gradient computation of GNNs, where the complexity is completely independent of the number of nodes, edges, and neighbors of the input. We proved its theoretical guarantee in terms of the approximation error. This is the first constant time approximation algorithm for GNNs in the literature. We further demonstrated that some existing GNNs cannot be approximated in constant time by any algorithm. Lastly, we validate the theory through experiments using synthetic and real-world datasets.

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

## A  MODELS

We introduce GraphSAGE-GCN, GraphSAGE-mean, GraphSAGE-pool, the graph convolutional networks (GCN), and the graph attention networks (GAT) for completeness of our paper.

**GraphSAGE-GCN** (Hamilton et al., 2017): The message function and the update function of this model is

$$M_{lvu}(\boldsymbol{z}_v, \boldsymbol{z}_u, \boldsymbol{e}_{vu}, \boldsymbol{\theta}) = \frac{\boldsymbol{z}_u}{\deg(v)},$$

$$U_l(\boldsymbol{z}_v, \boldsymbol{h}_v, \boldsymbol{\theta}) = \sigma(\boldsymbol{W}^{(l)}\boldsymbol{h}_v),$$

where $\boldsymbol{W}^{(l)}$ is a parameter matrix and $\sigma$ is an activation function such as sigmoid and ReLU. GraphSAGE-GCN includes the center node itself in the set of adjacent nodes (i.e., $\mathcal{N}(v) \leftarrow \mathcal{N}(v) \cup \{v\}$).

**GraphSAGE-mean** (Hamilton et al., 2017): The message function and the update function of this model is

$$M_{lvu}(\boldsymbol{z}_v, \boldsymbol{z}_u, \boldsymbol{e}_{vu}, \boldsymbol{\theta}) = \frac{\boldsymbol{z}_v}{\deg(v)},$$

$$U_l(\boldsymbol{z}_v, \boldsymbol{h}_v, \boldsymbol{\theta}) = \sigma(\boldsymbol{W}^{(l)}[\boldsymbol{z}_v, \boldsymbol{h}_v]),$$

where $[\cdot]$ denotes concatenation

**GraphSAGE-pool** (Hamilton et al., 2017): We do not formulate GraphSAGE-pool using the message function and the update function because it takes maximum instead of summation. The model of GraphSAGE-pool is

$$\boldsymbol{z}_v^{(l)} = \max(\{\sigma(\boldsymbol{W}^{(l)}\boldsymbol{z}_u^{(l-1)} + \boldsymbol{b}) \mid u \in \mathcal{N}(v)\}).$$

**GCN** (Kipf & Welling, 2017): The message function and the update function of this model is

$$M_{lvu}(\boldsymbol{z}_v, \boldsymbol{z}_u, \boldsymbol{e}_{vu}, \boldsymbol{\theta}) = \frac{\boldsymbol{z}^{(l)}}{\sqrt{\deg(v)\deg(u)}},$$

$$U_l(\boldsymbol{z}_v, \boldsymbol{h}_v, \boldsymbol{\theta}) = \sigma(\boldsymbol{W}^{(l)}\boldsymbol{h}_v).$$

**GAT** (Veličković et al., 2018): The message function and the update function of this model is

$$\alpha_{vu}^{(l)} = \frac{\exp(\text{LEAKYRELU}(\boldsymbol{a}^{(l)\top}[\boldsymbol{W}^{(l)}\boldsymbol{z}_v, \boldsymbol{W}^{(l)}\boldsymbol{z}_u]))}{\sum_{u' \in \mathcal{N}(v)} \exp(\text{LEAKYRELU}(\boldsymbol{a}^{(l)\top}[\boldsymbol{W}^{(l)}\boldsymbol{z}_v, \boldsymbol{W}^{(l)}\boldsymbol{z}_{u'}]))},$$

$$M_{lvu}(\boldsymbol{z}_v, \boldsymbol{z}_u, \boldsymbol{e}_{vu}, \boldsymbol{\theta}) = \alpha_{vu}^{(l)}\boldsymbol{z}_u,$$

$$U_l(\boldsymbol{z}_v, \boldsymbol{h}_v, \boldsymbol{\theta}) = \sigma(\boldsymbol{W}^{(l)}\boldsymbol{h}_v).$$

Note that, technically, MPNNs do not include GAT because GAT uses embeddings of other neighboring nodes to calculate the attention value $\alpha_{vu}$. However, we can apply the same argument as MPNNs to GAT, and neighbor sampling can approximate GAT in constant time as other MPNNs. To be precise, the following proposition holds true.

**Proposition 11.** *If Assumptions 1 holds true and $\sigma$ is Lipschitz continuous, if we take $r^{(L)} = O(\frac{1}{\varepsilon^2}\log\frac{1}{\delta})$ and $r^{(1)}, \ldots, r^{(L-1)} = O(\frac{1}{\varepsilon^2}(\log\frac{1}{\varepsilon} + \log\frac{1}{\delta}))$ samples, and let*

$$\hat{\boldsymbol{z}}_v^{(0)} = \boldsymbol{z}_v^{(0)}$$

$$\hat{\alpha}_{vu}^{(l)} = \frac{\exp(\text{LEAKYReLU}(\boldsymbol{a}^{(l)\top}[\boldsymbol{W}^{(l)}\hat{\boldsymbol{z}}_v^{(l-1)}, \boldsymbol{W}^{(l)}\hat{\boldsymbol{z}}_u^{(l-1)}]))}{\sum_{u' \in \mathcal{S}^{(l)}} \exp(\text{LEAKYReLU}(\boldsymbol{a}^{(l)\top}[\boldsymbol{W}^{(l)}\hat{\boldsymbol{z}}_v^{(l-1)}, \boldsymbol{W}^{(l)}\hat{\boldsymbol{z}}_{u'}^{(l-1)}]))},$$

$$\hat{\boldsymbol{h}}_v^{(l)} = \sum_{u \in \mathcal{S}^{(l)}} \hat{\alpha}_{vu}^{(l)} \hat{\boldsymbol{z}}_u^{(l-1)},$$

$$\hat{\boldsymbol{z}}_v^{(l)} = U_l(\hat{\boldsymbol{z}}_v^{(l-1)}, \hat{\boldsymbol{h}}_v^{(l)}, \boldsymbol{\theta}).$$

*Then, the following property holds true.*

$$\Pr[\|\boldsymbol{z}_v^{(L)} - \hat{\boldsymbol{z}}_v^{(L)}\|_2 \geq \varepsilon] < \delta.$$

We prove Proposition 11 in Section C.

# B EXPERIMENTAL SETUP

**Experiments for Q1:** For each algorithm and for $n = 2^7, 2^8, \ldots, 2^{15}$, we (1) run the algorithm with an input graph, (2) we calculate the exact embedding of one node and its estimate using 100 samples (i.e., $r^{(1)} = r^{(2)} = 100$), (3) calculate the approximation error. We run the process above 10 times, and we report the mean value of time consumption.

**Experiments for Q4:** We initialize the weight matrix $\boldsymbol{W}^{(1)}$ with normal distribution and then normalize it so that the operator norm $\|\boldsymbol{W}^{(1)}\|_{\text{op}}$ of the matrix equal to 1. This satisfies Assumption 1 (i.e., $\|\boldsymbol{\theta}\|_2 \leq \sqrt{2}$). For each $r = 1, \ldots, 10000$, we (1) initialize the weight matrix, (2) choose 400 nodes, (3) calculate the exact embedding of each chosen node, (4) calculate the estimate for each chosen node with $r$ samples (i.e., $r^{(1)} = r$), and (5) calculate the approximation error of each chosen node.

**Experiments for Q5:** In the experiment for the AB model, we (1) iterate $r$ from 8 to 1000, (2) set $n = r^2$, (3) generate 10 graphs with $n$ nodes using the BA model, (4) choose the node that has the maximum degree for each generated graph, (5) calculate the exact embeddings and its estimate for each chosen node with $r$ samples (i.e., $r^{(1)} = r^{(2)} = r$), and (6) calculate the approximation error.

The experimental process for the ER model is similar to that for the ER model, but we (1) use the ER model instead of BA model and (2) set $n = \text{floor}(r^{1.5})$ instead of $n = r^2$ to reduce the computational cost.

**Experiments for Q6:** To calculate the approximation errors of embeddings, for each trained model, for $r = 1 \ldots 100$, we (1) calculate the exact embedding of each test node, (2) calculate an estimate of embedding of each test node with $r$ samples (i.e., $r^{(1)} = r^{(2)} = r$), and (3) calculate the approximation error of each test node. To calculate the approximation errors of gradients, for each dataset, we (1) initialize ten models with Xavier initializer (Glorot & Bengio, 2010), (2) choose random 1000 nodes, and (3) for each model, for each chosen node, and for $r = 1 \ldots 100$, calculate the exact and approximation gradients of the classification loss with respect to the parameters, and we calculate their approximation error.

# C PROOFS

We introduce the following multivariate version of the Hoeffding's inequality (Hoeffding, 1963) to prove the theoretical bound (Theorem 1 and 4).

**Lemma 12** (multivariate Hoeffding's inequality). *Let $\boldsymbol{x}_1, \boldsymbol{x}_2, \ldots, \boldsymbol{x}_n$ be independent $d$-dimensional random variables whose two-norms are bounded $\|\boldsymbol{x}_i\|_2 \leq B$, and let $\bar{\boldsymbol{x}}$ be the empirical mean of these variables $\bar{\boldsymbol{x}} = \frac{1}{n}\sum_{i=1}^n \boldsymbol{x}_i$. Then, for any $\varepsilon > 0$,*

$$\Pr[\|\bar{\boldsymbol{x}} - \mathbb{E}[\bar{\boldsymbol{x}}]\|_2 \geq \varepsilon] \leq 2d \exp\left(-\frac{n\varepsilon^2}{2B^2 d}\right)$$

*holds true.*

Lemma 12 states that the empirical mean of $n = O(\frac{1}{\varepsilon^2} \log \frac{1}{\delta})$ samples independently sampled from the same distribution is the approximation of the exact mean with an absolute error of at most $\varepsilon$ and probability $1 - \delta$.

**Lemma 13** (Hoeffding's inequality (Hoeffding, 1963)). *Let $X_1, X_2, \ldots, X_n$ be independent random variables bounded by the intervals $[-B, B]$ and let $\bar{X}$ be the empirical mean of these variables $\bar{X} = \frac{1}{n} \sum_{i=1}^{n} X_i$. Then, for any $\varepsilon > 0$,*

$$\Pr[|\bar{X} - \mathbb{E}[\bar{X}]| \geq \varepsilon] \leq 2 \exp\left(-\frac{n\varepsilon^2}{2B^2}\right)$$

*holds true.*

*Proof of Lemma 12.* Apply Lemma 13 to each dimension $k$ of $X_i$. Then,

$$\Pr[|\bar{X}_k - \mathbb{E}[\bar{X}]_k| \geq \frac{\varepsilon}{\sqrt{d}}] \leq 2 \exp\left(-\frac{n\varepsilon^2}{2B^2 d}\right).$$

It should be noted that $|X_{ik}| < B$ because $\|X_i\|_2 < B$. Therefore,

$$\Pr[\exists k \in \{1, 2, \ldots, d\} \, |\bar{X}_k - \mathbb{E}[\bar{X}]_k| \geq \frac{\varepsilon}{\sqrt{d}}] \leq 2d \exp\left(-\frac{n\varepsilon^2}{2B^2 d}\right).$$

If $|\bar{X}_k - \mathbb{E}[\bar{X}]_k| < \frac{\varepsilon}{\sqrt{d}}$ holds true for all dimension $k$, then

$$\|\bar{X} - \mathbb{E}[\bar{X}]\|_2 = \sqrt{\sum_{k=1}^{d} (\bar{X}_k - \mathbb{E}[\bar{X}]_k)^2} < \sqrt{d \cdot \frac{\varepsilon^2}{d}} = \varepsilon.$$

Therefore,

$$\Pr[\|\bar{X} - \mathbb{E}[\bar{X}]\|_2 \geq \varepsilon] \leq 2d \exp\left(-\frac{n\varepsilon^2}{2B^2 d}\right).$$

$\square$

First, we prove that the embedding in each layer is bounded.

**Lemma 14.** *Under Assumptions 1 and 2, the norms of the embeddings $\|z_v^{(l)}\|_2$, $\|\hat{z}_v^{(l)}\|_2$, $\|h_v^{(l)}\|_2$, and $\|\hat{h}_v^{(l)}\|_2$ $(l = 1, \ldots, L)$ are bounded by a constant $B \in \mathbb{R}$.*

*Proof of Lemma 14.* We prove the theorem by performing mathematical induction. The norm of the input to the first layer is bounded by Assumption 1. The message function $\deg(i)M_{liu}$ and the update function $U_l$ is continuous by Assumption 2. Since the image $f(X)$ of a compact set $X \in \mathbb{R}^d$ is compact if $f$ is continuous, the images of $\deg(i)M_{liu}$ and $U_l$ are bounded by induction. $\square$

*Proof of Theorem 1.* We prove the theorem by performing mathematical induction on the number of layers $L$.

**Base case:** It is shown that the statement holds true for $L = 1$.

Because $U_L$ is uniform continuous,

$$\exists \varepsilon' > 0, \forall z_v, h_v, h_v', \theta, \|h_v - h_v'\|_2 < \varepsilon' \Rightarrow \|U_L(z_v, h_v, \theta) - U_L(z_v, h_v', \theta)\|_2 < \varepsilon. \quad (1)$$

Let $x_k$ be the $k$-th sample in $\mathcal{S}^{(L)}$ and $X_k = \deg(v)M_{Lvu}(z_v^{(0)}, z_{x_k}^{(0)}, e_{vx_k}, \theta)$. Then,

$$\mathbb{E}[X_k] = \sum_{u \in \mathcal{N}(v)} M_{Lvu}(z_v^{(0)}, z_u^{(0)}, e_{vu}, \theta) = h_v^{(L)}. \quad (2)$$

There exists a constant $C \in \mathbb{R}$ such that for any input satisfying Assumption 1,

$$\|X_k\|_2 < C \quad (3)$$

holds true because $\|z_v^{(0)}\|_2, \|z_{x_k}^{(0)}\|_2, \|e_{vx_k}\|_2$, and $\|\theta\|_2$ are bounded by Assumption 1 and $\deg(v)M_{Lvu}$ is continuous. Therefore, if we take $r^{(L)} = O(\frac{1}{\varepsilon'^2} \log \frac{1}{\delta})$ samples, $\Pr[\|\hat{h}_v^{(L)} - h_v^{(L)}\|_2 \geq \varepsilon'] \leq \delta$ by the Hoeffding's inequality and equations (2) and (3). Therefore, $\Pr[\|\hat{z}_v^{(L)} - z_v^{(L)}\|_2 \geq \varepsilon] \leq \delta$.

**Inductive step:** It is shown that the statement holds true for $L = l + 1$ if it holds true for $L = l$. The induction hypothesis is $\forall \varepsilon > 0, 1 > \delta > 0, \exists r^{(1)}(\varepsilon, \delta), \ldots, r^{(L-1)}(\varepsilon, \delta)$ such that $\forall v \in \mathcal{V}$, $\Pr[\|\hat{\mathcal{O}}_z^{(L-1)}(v, \varepsilon, \delta) - \boldsymbol{z}_v\|_2 \geq \varepsilon] \leq \delta$.

Because $U_L$ is uniform continuous,

$$\exists \varepsilon' > 0, \forall \boldsymbol{z}_v, \boldsymbol{h}_v, \boldsymbol{z}_v', \boldsymbol{h}_v', \boldsymbol{\theta}, \|[\boldsymbol{z}_v, \boldsymbol{h}_v] - [\boldsymbol{z}_v', \boldsymbol{h}_v']\|_2 < \varepsilon' \Rightarrow \|U_L(\boldsymbol{z}_v, \boldsymbol{h}_v, \boldsymbol{\theta}) - U_L(\boldsymbol{z}_v', \boldsymbol{h}_v', \boldsymbol{\theta})\|_2 < \varepsilon, \tag{4}$$

where $[\cdot]$ denotes concatenation. By the induction hypothesis,

$$\exists r'^{(1)}, \ldots, r'^{(L-1)} \text{ such that } \Pr[\|\hat{\mathcal{O}}_z^{(L-1)}(v) - \boldsymbol{z}_v^{(L-1)}\|_2 \geq \varepsilon'/\sqrt{2}] \leq \delta/2. \tag{5}$$

holds true. Let

$$\tilde{\boldsymbol{h}}_v^{(L)} = \frac{\deg(v)}{r^{(L)}} \sum_{u \in \mathcal{S}^{(L)}} M_{Lvu}(\boldsymbol{z}_v^{(L-1)}, \boldsymbol{z}_u^{(L-1)}, \boldsymbol{e}_{vu}, \boldsymbol{\theta}).$$

Let $x_k$ be the $k$-th sample in $\mathcal{S}^{(L)}$ and $X_k = \deg(v) M_{Lvu}(\boldsymbol{z}_v^{(L-1)}, z_{x_k}^{(L-1)}, \boldsymbol{e}_{vx_k}, \boldsymbol{\theta})$. Then,

$$\mathbb{E}[X_k] = \sum_{u \in \mathcal{N}(v)} M_{Lvu}(\boldsymbol{z}_v^{(L-1)}, \boldsymbol{z}_{x_k}^{(L-1)}, \boldsymbol{e}_{vx_k}, \boldsymbol{\theta}) = \boldsymbol{h}_v^{(L)}. \tag{6}$$

There exists a constant $C \in \mathbb{R}$ such that for any input satisfying Assumption 1,

$$\|X_k\|_2 < C, \tag{7}$$

because $\|\boldsymbol{z}_v^{(L-1)}\|_2, \|z_{x_k}^{(L-1)}\|_2, \|\boldsymbol{e}_{vx_k}\|_2$, and $\|\boldsymbol{\theta}\|_2$ are bounded by Assumption 1 and Theorem 14, and $\deg(v) M_{Lvu}$ is continuous. If we take $r^{(L)} = O(\frac{1}{\varepsilon'^2} \log \frac{1}{\delta})$, then

$$\Pr[\|\tilde{\boldsymbol{h}}_v^{(l)} - \boldsymbol{h}_v^{(l)}\|_2 \geq \varepsilon'/(2\sqrt{2})] \leq \delta/4, \tag{8}$$

by the Hoeffding's inequality and equations (6) and (7). Because $\deg(v) M_{Lvu}$ is uniform continuous,

$$\exists \varepsilon'' > 0 \text{ such that } \|[\boldsymbol{z}_v^{(L-1)}, \boldsymbol{z}_u^{(L-1)}] - [\boldsymbol{z}_v'^{(L-1)}, \boldsymbol{z}_u'^{(L-1)}]\|_2 \leq \varepsilon''$$
$$\Rightarrow \deg(v) \| M_{Lvu}(\boldsymbol{z}_v^{(L-1)}, \boldsymbol{z}_u^{(L-1)}, \boldsymbol{e}_{vu}, \boldsymbol{\theta}) - M_{Lvu}(\boldsymbol{z}_v'^{(L-1)}, \boldsymbol{z}_u'^{(L-1)}, \boldsymbol{e}_{vu}, \boldsymbol{\theta})\|_2 \leq \varepsilon'/(2\sqrt{2}). \tag{9}$$

By the induction hypothesis,

$$\exists r''^{(1)}, \ldots, r''^{(l)} \text{ such that } \Pr[\|\hat{\mathcal{O}}_z^{(L-1)}(v) - \boldsymbol{z}_v^{(L-1)}\|_2 \geq \varepsilon''/\sqrt{2}] \leq \delta/(8r^{(L)}). \tag{10}$$

Therefore, the probability that the errors of all oracle calls are bounded is

$$\Pr[\exists u \in \mathcal{S}^{(L)}, \|[\hat{\mathcal{O}}_z^{(L-1)}(v), \hat{\mathcal{O}}_z^{(L-1)}(u)] - [\boldsymbol{z}_v^{(L-1)}, \boldsymbol{z}_u^{(L-1)}]\|_2 \geq \varepsilon''] \leq \delta/4. \tag{11}$$

By equations (9) and (11),

$\Pr[\exists u \in \mathcal{S}^{(L)},$

$\deg(v) \| M_{Lvu}(\boldsymbol{z}_v^{(L-1)}, \boldsymbol{z}_u^{(L-1)}, \boldsymbol{e}_{vu}, \boldsymbol{\theta}) - M_{Lvu}(\hat{\boldsymbol{z}}_v^{(L-1)}, \hat{\boldsymbol{z}}_u^{(L-1)}, \boldsymbol{e}_{vu}, \boldsymbol{\theta})\|_2 \geq \varepsilon'/(2\sqrt{2})] \leq \delta/4.$

$$\Pr[\|\tilde{\boldsymbol{h}}_v^{(L)} - \hat{\boldsymbol{h}}_v^{(L)}\|_2 \geq \varepsilon'/(2\sqrt{2})] \leq \delta/4. \tag{12}$$

By the triangular inequality and equations (8) and (12),

$$\Pr[\|\hat{\boldsymbol{h}}_v^{(L)} - \boldsymbol{h}_v^{(L)}\|_2 \geq \varepsilon'/\sqrt{2}] \leq \delta/2. \tag{13}$$

Therefore, if we take $r^{(1)} = \max(r'^{(1)}, r''^{(1)}), \ldots, r^{(L-1)} = \max(r'^{(L-1)}, r''^{(L-1)})$, by equations (5) and (13),

$$\Pr[\|[\boldsymbol{z}_v^{(L-1)}, \boldsymbol{h}_v^{(L)}] - [\hat{\mathcal{O}}_z^{(L-1)}(v), \hat{\boldsymbol{h}}_v^{(L)}]\|_2 \geq \varepsilon'] \leq \delta. \tag{14}$$

Therefore, by equations (9) and (14),

$$\Pr[\|\hat{\boldsymbol{z}}_v^{(L)} - \boldsymbol{z}_v^{(L)}\|_2 \geq \varepsilon] \leq \delta.$$

$\square$

*Proof of Theorem 2.* We prove this by performing mathematical induction on the number of layers.

**Base case:** It is shown that the statement holds true for $L = 1$.

If $U_L$ is $K$-Lipschitz continuous, $\varepsilon' = O(\varepsilon)$ in equation (1). Therefore, $r^{(L)} = O(\frac{1}{\varepsilon^2} \log \frac{1}{\delta})$.

**Inductive step:** It is shown that the statement holds true for $L = l + 1$ if it holds true for $L = l$.

If $U_L$ and $M_{Lvu}$ are $K$-Lipschitz continuous, $\varepsilon' = O(\varepsilon)$ in equation (4) and $\varepsilon'' = O(\varepsilon)$ in equation (9). Therefore, $r^{(L)} = O(\frac{1}{\varepsilon^2} \log \frac{1}{\delta})$. We call $\hat{\mathcal{O}}_z^{(L-1)}(v)$ such that $\Pr[\|\hat{\mathcal{O}}_z^{(L-1)}(v) - \boldsymbol{z}_v^{(L-1)}\|_2 \geq \varepsilon'/\sqrt{2}] \leq \delta/2$ in equation (5). Therefore, $r'^{(1)}, \ldots, r'^{(L-1)} = O(\frac{1}{\varepsilon^2}(\log \frac{1}{\varepsilon} + \log \frac{1}{\delta})$ are sufficient by the induction hypothesis. We call $\hat{\mathcal{O}}_z^{(L-1)}(v)$ such that $\Pr[\|\hat{\mathcal{O}}_z^{(L-1)}(v) - \boldsymbol{z}_v^{(L-1)}\|_2 \geq \varepsilon''/\sqrt{2}] \leq \delta/(8r^{(L)})$ in equation (10). Therefore, $r''^{(1)}, \ldots, r''^{(L-1)} = O(\frac{1}{\varepsilon^2}(\log \frac{1}{\varepsilon} + \log \frac{1}{\delta})$ are sufficient by the induction hypothesis because $\log \frac{1}{\delta/(8r^{(L)})} = O(\log \frac{1}{\varepsilon} + \log \frac{1}{\delta})$. In total, the complexity is $O(\frac{1}{\varepsilon^{2L}}(\log \frac{1}{\varepsilon} + \log \frac{1}{\varepsilon})^{L-1} \log \frac{1}{\delta})$ □

**Lemma 15** (Chazelle et al. (2005)). *Let $\mathcal{D}^s$ be $Bernoulli(\frac{1+s\varepsilon}{2})$. Let $n$-dimentional distribution $\mathcal{D}$ be (1) pick $s = 1$ with probability $1/2$ and $s = -1$ otherwise; (2) then draw $n$ values from $\mathcal{D}^s$. Any probabilistic algorithm that can guess the value of $s$ with a probability error below $1/4$ requires $\Omega(\frac{1}{\varepsilon^2})$ bit lookup on average.*

*Proof of Theorem 3.* We prove there is a counter example in the GraphSAGE-GCN models. Suppose there is an algorithm that approximates the one-layer GraphSAGE-GCN within $o(\varepsilon^2)$ queries. We prove that this algorithm can distinguish $\mathcal{D}$ in Lemma 15 within $o(\varepsilon^2)$ queries and derive a contradiction.

Let $\sigma$ be any non-constant $K$-Lipschitz activation function. There exists $a, b \in \mathbb{R}$ $(a > b)$ such that $\sigma(a) \neq \sigma(b)$ because $\sigma$ is not constant. Let $S = \frac{|\sigma(a) - \sigma(b)|}{a - b} > 0$. Let $\varepsilon > 0$ be any sufficiently small positive value and $t \in \{0, 1\}^n$ be a random variable drawn from $\mathcal{D}$. We prove that we can determine $s$ with high provability within $o(\varepsilon^2)$ queries using the algorithm. Let $G$ be a clique $K_n$ and $\boldsymbol{W}^{(1)} = 1$. Let us calculate $a_\varepsilon$ and $b_\varepsilon$ using the following steps: (1) set $a_\varepsilon = a$ and $b_\varepsilon = b$; (2) if $a_\varepsilon - b_\varepsilon < \varepsilon$, return $a_\varepsilon$ and $b_\varepsilon$; (3) $m = \frac{a_\varepsilon + b_\varepsilon}{2}$; (4) if $|\sigma(a_\varepsilon) - \sigma(m)| > |\sigma(m) - \sigma(b_\varepsilon)|$, then set $b_\varepsilon = m$, otherwise $a_\varepsilon = m$; and (5) go back to (2). Here, $\varepsilon/2 \leq a_\varepsilon - b_\varepsilon < \varepsilon$, $a \leq \frac{a_\varepsilon + b_\varepsilon}{2} \leq b$, and $|\sigma(a_\varepsilon) - \sigma(b_\varepsilon)| \geq \frac{S}{2}\varepsilon$ hold true. Let $x_v = \frac{a_\varepsilon + b_\varepsilon}{2} + (2t_v - 1)\frac{a_\varepsilon - b_\varepsilon}{2\varepsilon}$ for all $v \in \mathcal{V}$. Then, $\mathbb{E}[h_v \mid s = 1] = a_\varepsilon$ and $\mathbb{E}[h_v \mid s = -1] = b_\varepsilon$. Therefore, $\Pr[|z_v - \sigma(a_\varepsilon)| < \frac{S}{8}\varepsilon \mid s = 1] \to 1$ as $n \to \infty$ and $\Pr[|z_v - \sigma(b_\varepsilon)| < \frac{S}{8}\varepsilon \mid s = -1] \to 1$ as $n \to \infty$ because $\sigma$ is $K$-Lipschitz. We set the error tolerance to $\frac{S}{8}\varepsilon$ and $n$ to a sufficiently large number. Then $s = 1$ if $|\hat{z}_v - \sigma(a_\varepsilon)| < \frac{S}{4}\varepsilon$ and $s = -1$ otherwise with high probability. However, the algorithm accesses $t$ (i.e., accesses $\hat{\mathcal{O}}_{\text{feature}}$) $o(\varepsilon^2)$ times. This contradicts with Lemma 15. □

**Lemma 16.** *Under Assumptions 1, 2 and 3, the norms of the gradients of the message functions and the update functions $\|DU_l(\boldsymbol{z}_v^{(l-1)}, \boldsymbol{h}_v^{(l)}, \boldsymbol{\theta})\|_F$, $\|DU_l(\hat{\boldsymbol{z}}_v^{(l-1)}, \hat{\boldsymbol{h}}_v^{(l)}, \boldsymbol{\theta})\|_F$, $\|deg(v)DM_{lvu}(\boldsymbol{z}_v^{(l-1)}, \boldsymbol{v}_u^{(l)}, \boldsymbol{e}_{vu}, \boldsymbol{\theta})\|_F$, and $\|deg(v)DM_{lvu}(\hat{\boldsymbol{z}}_v^{(l-1)}, \hat{\boldsymbol{v}}_u^{(l)}, \boldsymbol{e}_{vu}, \boldsymbol{\theta})\|_F$ are bounded by a constant $B' \in \mathbb{R}$.*

*Proof of Lemma 16.* The input of each function is bounded by Lemma 14. Because $DU_l$ and $deg(v)DM_{lvu}$ is uniform continuous, these images are bounded. □

*Proof of Theorem 4.* We prove the theorem by performing mathematical induction on the number of layers $L$.

**Base case:** It is shown that the statement holds true for $L = 1$.

When the number of layers is one,

$$\frac{\partial \boldsymbol{z}_v^{(L)}}{\partial \boldsymbol{\theta}} = \frac{\partial U_L}{\partial \boldsymbol{\theta}}(\boldsymbol{z}_v^{(0)}, \boldsymbol{h}_v^{(L)}, \boldsymbol{\theta}) + \frac{\partial U_L}{\partial \boldsymbol{h}_v^{(L)}}(\boldsymbol{z}_v^{(0)}, \boldsymbol{h}_v^{(L)}, \boldsymbol{\theta}) \frac{\partial \boldsymbol{h}_v^{(L)}}{\partial \boldsymbol{\theta}}$$

$$= \frac{\partial U_L}{\partial \boldsymbol{\theta}}(\boldsymbol{z}_v^{(0)}, \boldsymbol{h}_v^{(L)}, \boldsymbol{\theta}) + \frac{\partial U_L}{\partial \boldsymbol{h}_v^{(L)}}(\boldsymbol{z}_v^{(0)}, \boldsymbol{h}_v^{(L)}, \boldsymbol{\theta}) \sum_{u \in \mathcal{N}(v)} \frac{\partial M_{Lvu}}{\partial \boldsymbol{\theta}}(\boldsymbol{z}_v^{(0)}, \boldsymbol{z}_u^{(0)}, \boldsymbol{e}_{vu}, \boldsymbol{\theta}).$$

$$\frac{\partial \hat{\boldsymbol{z}}_v^{(L)}}{\partial \boldsymbol{\theta}} = \frac{\partial U_L}{\partial \boldsymbol{\theta}}(\boldsymbol{z}_v^{(0)}, \hat{\boldsymbol{h}}_v^{(L)}, \boldsymbol{\theta}) + \frac{\partial U_L}{\partial \hat{\boldsymbol{h}}_v^{(L)}}(\boldsymbol{z}_v^{(0)}, \hat{\boldsymbol{h}}_v^{(L)}, \boldsymbol{\theta}) \frac{\partial \hat{\boldsymbol{h}}_v^{(L)}}{\partial \boldsymbol{\theta}}$$

$$= \frac{\partial U_L}{\partial \boldsymbol{\theta}}(\boldsymbol{z}_v^{(0)}, \hat{\boldsymbol{h}}_v^{(L)}, \boldsymbol{\theta}) + \frac{\partial U_L}{\partial \hat{\boldsymbol{h}}_v^{(L)}}(\boldsymbol{z}_v^{(0)}, \hat{\boldsymbol{h}}_v^{(L)}, \boldsymbol{\theta}) \frac{\deg(v)}{r^{(L)}} \sum_{u \in \mathcal{S}^{(L)}} \frac{\partial M_{Lvu}}{\partial \boldsymbol{\theta}}(\boldsymbol{z}_v^{(0)}, \boldsymbol{z}_u^{(0)}, \boldsymbol{e}_{vu}, \boldsymbol{\theta}).$$

Because $DU_L$ is uniform continuous,

$$\exists \varepsilon' > 0 \text{ s.t. for all input}, \|\boldsymbol{h}_v^{(L)} - \hat{\boldsymbol{h}}^{(L)}\|_2 < \varepsilon' \Rightarrow$$
$$(\|\frac{\partial U_L}{\partial \boldsymbol{\theta}}(\boldsymbol{z}_v^{(0)}, \boldsymbol{h}_v^{(L)}, \boldsymbol{\theta}) - \frac{\partial U_L}{\partial \boldsymbol{\theta}}(\boldsymbol{z}_v^{(0)}, \hat{\boldsymbol{h}}_v^{(L)}, \boldsymbol{\theta})\|_F < \varepsilon/2 \wedge$$
$$\|\frac{\partial U_L}{\partial \boldsymbol{h}_v^{(L)}}(\boldsymbol{z}_v^{(0)}, \boldsymbol{h}_v^{(L)}, \boldsymbol{\theta}) - \frac{\partial U_L}{\partial \hat{\boldsymbol{h}}_v^{(L)}}(\boldsymbol{z}_v^{(0)}, \hat{\boldsymbol{h}}_v^{(L)}, \boldsymbol{\theta})\|_F < \varepsilon/(4B')). \tag{15}$$

If we take $r^{(L)} = O(\frac{1}{\varepsilon'^2} \log \frac{1}{\delta})$,

$$\Pr[\|\boldsymbol{h}_v^{(L)} - \hat{\boldsymbol{h}}^{(L)}\|_2 \geq \varepsilon'] \leq \delta/2 \tag{16}$$

holds true for any input by the argument of the proof of Theorem 1.

Let $x_k$ be the $k$-th sample in $\mathcal{S}^{(L)}$ and $X_k = \deg(v)\frac{\partial M_{Lvu}}{\partial \boldsymbol{\theta}}(\boldsymbol{z}_v^{(0)}, \boldsymbol{z}_u^{(0)}, \boldsymbol{e}_{vu}, \boldsymbol{\theta})$. Then,

$$\mathbb{E}[X_k] = \sum_{u \in \mathcal{N}(v)} \frac{\partial M_{Lvu}}{\partial \boldsymbol{\theta}}(\boldsymbol{z}_v^{(0)}, \boldsymbol{z}_u^{(0)}, \boldsymbol{e}_{vu}, \boldsymbol{\theta}) = \frac{\partial \boldsymbol{h}_v^{(L)}}{\partial \boldsymbol{\theta}}. \tag{17}$$

There exists a constant $C \in \mathbb{R}$ such that for any input satisfying Assumption 1,

$$\|X_k\|_2 < C, \tag{18}$$

because $\|\boldsymbol{z}_v^{(0)}\|_2, \|\boldsymbol{z}_{x_k}^{(0)}\|_2, \|\boldsymbol{e}_{vx_k}\|_2$, and $\|\boldsymbol{\theta}\|_2$ are bounded by Assumption 1 and $\deg(v)DM_{Lvu}$ is continuous. Therefore, if we take $r^{(L)} = O(\frac{1}{\varepsilon^2} \log \frac{1}{\delta})$ samples,

$$\Pr[\|\frac{\partial \hat{\boldsymbol{h}}_v^{(L)}}{\partial \boldsymbol{\theta}} - \frac{\partial \boldsymbol{h}_v^{(L)}}{\partial \boldsymbol{\theta}}\|_F \geq \varepsilon/(4B')] \leq \delta/2 \tag{19}$$

holds true by the Hoeffding's inequality and equations (17) and (18). If

$$\|\frac{\partial \hat{\boldsymbol{h}}_v^{(L)}}{\partial \boldsymbol{\theta}} - \frac{\partial \boldsymbol{h}_v^{(L)}}{\partial \boldsymbol{\theta}}\|_F < \varepsilon/(4B')$$

and

$$\|\frac{\partial U_L}{\partial \boldsymbol{h}_v^{(L)}}(\boldsymbol{z}_v^{(0)}, \boldsymbol{h}_v^{(L)}, \boldsymbol{\theta}) - \frac{\partial U_L}{\partial \hat{\boldsymbol{h}}_v^{(L)}}(\boldsymbol{z}_v^{(0)}, \hat{\boldsymbol{h}}_v^{(L)}, \boldsymbol{\theta})\|_F < \varepsilon/(4B'))$$

hold true, then

$$\|\frac{\partial \hat{\boldsymbol{h}}_v^{(L)}}{\partial \boldsymbol{\theta}} \frac{\partial U_L}{\partial \hat{\boldsymbol{h}}_v^{(L)}}(\boldsymbol{z}_v^{(0)}, \hat{\boldsymbol{h}}_v^{(L)}, \boldsymbol{\theta}) - \frac{\partial \boldsymbol{h}_v^{(L)}}{\partial \boldsymbol{\theta}} \frac{\partial U_L}{\partial \boldsymbol{h}_v^{(L)}}(\boldsymbol{z}_v^{(0)}, \boldsymbol{h}_v^{(L)}, \boldsymbol{\theta})\|_F$$

$$\leq \|\frac{\partial \hat{\boldsymbol{h}}_v^{(L)}}{\partial \boldsymbol{\theta}} \frac{\partial U_L}{\partial \hat{\boldsymbol{h}}_v^{(L)}}(\boldsymbol{z}_v^{(0)}, \hat{\boldsymbol{h}}_v^{(L)}, \boldsymbol{\theta}) - \frac{\partial \hat{\boldsymbol{h}}_v^{(L)}}{\partial \boldsymbol{\theta}} \frac{\partial U_L}{\partial \boldsymbol{h}_v^{(L)}}(\boldsymbol{z}_v^{(0)}, \boldsymbol{h}_v^{(L)}, \boldsymbol{\theta})\|_F$$

$$+ \|\frac{\partial \hat{\boldsymbol{h}}_v^{(L)}}{\partial \boldsymbol{\theta}} \frac{\partial U_L}{\partial \boldsymbol{h}_v^{(L)}}(\boldsymbol{z}_v^{(0)}, \boldsymbol{h}_v^{(L)}, \boldsymbol{\theta}) - \frac{\partial \boldsymbol{h}_v^{(L)}}{\partial \boldsymbol{\theta}} \frac{\partial U_L}{\partial \boldsymbol{h}_v^{(L)}}(\boldsymbol{z}_v^{(0)}, \boldsymbol{h}_v^{(L)}, \boldsymbol{\theta})\|_F$$

$$= \|\frac{\partial \hat{\boldsymbol{h}}_v^{(L)}}{\partial \boldsymbol{\theta}}\|_F \|\frac{\partial U_L}{\partial \hat{\boldsymbol{h}}_v^{(L)}}(\boldsymbol{z}_v^{(0)}, \hat{\boldsymbol{h}}_v^{(L)}, \boldsymbol{\theta}) - \frac{\partial U_L}{\partial \boldsymbol{h}_v^{(L)}}(\boldsymbol{z}_v^{(0)}, \boldsymbol{h}_v^{(L)}, \boldsymbol{\theta})\|_F$$

$$+ \|\frac{\partial \hat{\boldsymbol{h}}_v^{(L)}}{\partial \boldsymbol{\theta}} - \frac{\partial \boldsymbol{h}_v^{(L)}}{\partial \boldsymbol{\theta}}\|_F \|\frac{\partial U_L}{\partial \boldsymbol{h}_v^{(L)}}(\boldsymbol{z}_v^{(0)}, \boldsymbol{h}_v^{(L)}, \boldsymbol{\theta})\|_F$$

$$\leq B' \|\frac{\partial U_L}{\partial \hat{\boldsymbol{h}}_v^{(L)}}(\boldsymbol{z}_v^{(0)}, \hat{\boldsymbol{h}}_v^{(L)}, \boldsymbol{\theta}) - \frac{\partial U_L}{\partial \boldsymbol{h}_v^{(L)}}(\boldsymbol{z}_v^{(0)}, \boldsymbol{h}_v^{(L)}, \boldsymbol{\theta})\|_F + B' \|\frac{\partial \hat{\boldsymbol{h}}_v^{(L)}}{\partial \boldsymbol{\theta}} - \frac{\partial \boldsymbol{h}_v^{(L)}}{\partial \boldsymbol{\theta}}\|_F$$

$$< B' \frac{\varepsilon}{4B'} + B' \frac{\varepsilon}{4B'} = \frac{\varepsilon}{2}.$$

Therefore, $\Pr[\|\frac{\partial \boldsymbol{z}_v^{(L)}}{\partial \boldsymbol{\theta}} - \frac{\partial \boldsymbol{z}_v^{(L)}}{\partial \boldsymbol{\theta}}\|_F \geq \varepsilon] \leq \delta$ by equations (15), (16), and (19).

**Inductive step:** It is shown that the statement holds true for $L = l + 1$ if it holds true for $L = l$.

$$\frac{\partial \boldsymbol{z}_v^{(L)}}{\partial \boldsymbol{\theta}} = \frac{\partial U_L}{\partial \boldsymbol{\theta}}(\boldsymbol{z}_v^{(L-1)}, \boldsymbol{h}_v^{(L)}, \boldsymbol{\theta}) + \frac{\partial U_L}{\partial \boldsymbol{z}_v^{(L-1)}}(\boldsymbol{z}_v^{(L-1)}, \boldsymbol{h}_v^{(L)}, \boldsymbol{\theta}) \frac{\partial \boldsymbol{z}_v^{(L-1)}}{\partial \boldsymbol{\theta}}$$

$$+ \frac{\partial U_L}{\partial \boldsymbol{h}_v^{(L)}}(\boldsymbol{z}_v^{(L-1)}, \boldsymbol{h}_v^{(L)}, \boldsymbol{\theta}) \sum_{u \in \mathcal{N}(v)} \frac{\partial M_{Lvu}}{\partial \boldsymbol{\theta}}(\boldsymbol{z}_v^{(L-1)}, \boldsymbol{z}_u^{(L-1)}, \boldsymbol{e}_{vu}, \boldsymbol{\theta})$$

$$+ \frac{\partial U_L}{\partial \boldsymbol{h}_v^{(L)}}(\boldsymbol{z}_v^{(L-1)}, \boldsymbol{h}_v^{(L)}, \boldsymbol{\theta}) \sum_{u \in \mathcal{N}(v)} \frac{\partial M_{Lvu}}{\partial \boldsymbol{z}_v^{(L-1)}}(\boldsymbol{z}_v^{(L-1)}, \boldsymbol{z}_u^{(L-1)}, \boldsymbol{e}_{vu}, \boldsymbol{\theta}) \frac{\partial \boldsymbol{z}_v^{(L-1)}}{\partial \boldsymbol{\theta}}$$

$$+ \frac{\partial U_L}{\partial \boldsymbol{h}_v^{(L)}}(\boldsymbol{z}_v^{(L-1)}, \boldsymbol{h}_v^{(L)}, \boldsymbol{\theta}) \sum_{u \in \mathcal{N}(v)} \frac{\partial M_{Lvu}}{\partial \boldsymbol{z}_u^{(L-1)}}(\boldsymbol{z}_v^{(L-1)}, \boldsymbol{z}_u^{(L-1)}, \boldsymbol{e}_{vu}, \boldsymbol{\theta}) \frac{\partial \boldsymbol{z}_u^{(L-1)}}{\partial \boldsymbol{\theta}}.$$

$$\frac{\partial \hat{\boldsymbol{z}}_v^{(L)}}{\partial \boldsymbol{\theta}} = \frac{\partial U_L}{\partial \boldsymbol{\theta}}(\hat{\boldsymbol{z}}_v^{(L-1)}, \hat{\boldsymbol{h}}_v^{(L)}, \boldsymbol{\theta}) + \frac{\partial U_L}{\partial \hat{\boldsymbol{z}}_v^{(L-1)}}(\hat{\boldsymbol{z}}_v^{(L-1)}, \hat{\boldsymbol{h}}_v^{(L)}, \boldsymbol{\theta}) \frac{\partial \hat{\boldsymbol{z}}_v^{(L-1)}}{\partial \boldsymbol{\theta}}$$

$$+ \frac{\partial U_L}{\partial \hat{\boldsymbol{h}}_v^{(L)}}(\hat{\boldsymbol{z}}_v^{(L-1)}, \hat{\boldsymbol{h}}_v^{(L)}, \boldsymbol{\theta}) \frac{\deg(v)}{r^{(L)}} \sum_{u \in \mathcal{S}^{(L)}} \frac{\partial M_{Lvu}}{\partial \boldsymbol{\theta}}(\hat{\boldsymbol{z}}_v^{(L-1)}, \hat{\boldsymbol{z}}_u^{(L-1)}, \boldsymbol{e}_{vu}, \boldsymbol{\theta})$$

$$+ \frac{\partial U_L}{\partial \hat{\boldsymbol{h}}_v^{(L)}}(\hat{\boldsymbol{z}}_v^{(L-1)}, \hat{\boldsymbol{h}}_v^{(L)}, \boldsymbol{\theta}) \frac{\deg(v)}{r^{(L)}} \sum_{u \in \mathcal{S}^{(L)}} \frac{\partial M_{Lvu}}{\partial \hat{\boldsymbol{z}}_v^{(L-1)}}(\hat{\boldsymbol{z}}_v^{(L-1)}, \hat{\boldsymbol{z}}_u^{(L-1)}, \boldsymbol{e}_{vu}, \boldsymbol{\theta}) \frac{\partial \hat{\boldsymbol{z}}_v^{(L-1)}}{\partial \boldsymbol{\theta}}$$

$$+ \frac{\partial U_L}{\partial \hat{\boldsymbol{h}}_v^{(L)}}(\hat{\boldsymbol{z}}_v^{(L-1)}, \hat{\boldsymbol{h}}_v^{(L)}, \boldsymbol{\theta}) \frac{\deg(v)}{r^{(L)}} \sum_{u \in \mathcal{S}^{(L)}} \frac{\partial M_{Lvu}}{\partial \hat{\boldsymbol{z}}_u^{(L-1)}}(\hat{\boldsymbol{z}}_v^{(L-1)}, \hat{\boldsymbol{z}}_u^{(L-1)}, \boldsymbol{e}_{vu}, \boldsymbol{\theta}) \frac{\partial \hat{\boldsymbol{z}}_u^{(L-1)}}{\partial \boldsymbol{\theta}}.$$

Because $DU_L$ is uniform continuous,

$$\exists \varepsilon' > 0 \text{ s.t. for all input,} \|[\boldsymbol{z}_v^{(L-1)}, \boldsymbol{h}_v^{(L)}] - [\hat{\boldsymbol{z}}_v^{(L-1)}, \hat{\boldsymbol{h}}_v^{(L)}]\|_2 < \varepsilon' \Rightarrow$$

$$(\|\frac{\partial U_L}{\partial \boldsymbol{\theta}}(\boldsymbol{z}_v^{(L-1)}, \boldsymbol{h}_v^{(L)}, \boldsymbol{\theta}) - \frac{\partial U_L}{\partial \boldsymbol{\theta}}(\hat{\boldsymbol{z}}_v^{(L-1)}, \hat{\boldsymbol{h}}_v^{(L)}, \boldsymbol{\theta})\|_F < O(\varepsilon) \wedge$$

$$\|\frac{\partial U_L}{\partial \boldsymbol{h}_v^{(L)}}(\boldsymbol{z}_v^{(L-1)}, \boldsymbol{h}_v^{(L)}, \boldsymbol{\theta}) - \frac{\partial U_L}{\partial \hat{\boldsymbol{h}}_v^{(L)}}(\hat{\boldsymbol{z}}_v^{(L-1)}, \hat{\boldsymbol{h}}_v^{(L)}, \boldsymbol{\theta})\|_F < O(\varepsilon) \wedge$$

$$\|\frac{\partial U_L}{\partial \boldsymbol{z}_v^{(L-1)}}(\boldsymbol{z}_v^{(L-1)}, \boldsymbol{h}_v^{(L)}, \boldsymbol{\theta}) - \frac{\partial U_L}{\partial \hat{\boldsymbol{z}}_v^{(L-1)}}(\hat{\boldsymbol{z}}_v^{(L-1)}, \hat{\boldsymbol{h}}_v^{(L)}, \boldsymbol{\theta})\|_F < O(\varepsilon)). \tag{20}$$

Because $\deg(v)DM_{Lvu}$ is uniform continuous,

$$\exists \varepsilon'' > 0 \text{ s.t. for all input,} \|[\boldsymbol{z}_v^{(L-1)}, \boldsymbol{z}_u^{(L-1)}] - [\hat{\boldsymbol{z}}_v^{(L-1)}, \hat{\boldsymbol{z}}_u^{(L-1)}]\|_2 < \varepsilon'' \Rightarrow$$

$$(\deg(v)\|\frac{\partial M_{Lvu}}{\partial \boldsymbol{\theta}}(\boldsymbol{z}_v^{(L-1)}, \boldsymbol{z}_u^{(L-1)}, \boldsymbol{e}_{vu}, \boldsymbol{\theta}) - \frac{\partial U_L}{\partial \boldsymbol{\theta}}(\hat{\boldsymbol{z}}_v^{(L-1)}, \hat{\boldsymbol{z}}_u^{(L-1)}, \boldsymbol{e}_{vu}, \boldsymbol{\theta})\|_F < O(\varepsilon) \wedge$$

$$\deg(v)\|\frac{\partial M_{Lvu}}{\partial \boldsymbol{h}_v^{(L)}}(\boldsymbol{z}_v^{(L-1)}, \boldsymbol{z}_u^{(L-1)}, \boldsymbol{e}_{vu}, \boldsymbol{\theta}) - \frac{\partial U_L}{\partial \hat{\boldsymbol{h}}_v^{(L)}}(\hat{\boldsymbol{z}}_v^{(L-1)}, \hat{\boldsymbol{z}}_u^{(L-1)}, \boldsymbol{e}_{vu}, \boldsymbol{\theta})\|_F < O(\varepsilon) \wedge$$

$$\deg(v)\|\frac{\partial M_{Lvu}}{\partial \boldsymbol{z}_v^{(L-1)}}(\boldsymbol{z}_v^{(L-1)}, \boldsymbol{z}_u^{(L-1)}, \boldsymbol{e}_{vu}, \boldsymbol{\theta}) - \frac{\partial U_L}{\partial \hat{\boldsymbol{z}}_v^{(L-1)}}(\hat{\boldsymbol{z}}_v^{(L-1)}, \hat{\boldsymbol{z}}_u^{(L-1)}, \boldsymbol{e}_{vu}, \boldsymbol{\theta})\|_F < O(\varepsilon)). \tag{21}$$

By the argument of the proof of Theorem 1, if we take sufficiently large number of samples,

$$\Pr[\|\boldsymbol{z}_v^{(L-1)} - \hat{\boldsymbol{z}}_v^{(L-1)}\|_2 \geq O(\min(\varepsilon', \varepsilon''))] \leq O(\varepsilon\delta), \tag{22}$$

$$\Pr[\|\boldsymbol{h}_v^{(L)} - \hat{\boldsymbol{h}}_v^{(L)}\|_2 \geq O(\varepsilon')] \leq O(\delta). \tag{23}$$

By the induction hypothesis, there exists $r^{(1)}, \ldots, r^{(L-1)}$ such that

$$\Pr[\|\frac{\partial \boldsymbol{z}_u^{(L-1)}}{\partial \boldsymbol{\theta}} - \frac{\partial \hat{\boldsymbol{z}}_u^{(L-1)}}{\partial \boldsymbol{\theta}}\|_F \geq O(\varepsilon)] \leq O(\varepsilon\delta),$$

$$\Pr[\|\frac{\partial \boldsymbol{z}_v^{(L-1)}}{\partial \boldsymbol{\theta}} - \frac{\partial \hat{\boldsymbol{z}}_v^{(L-1)}}{\partial \boldsymbol{\theta}}\|_F \geq O(\varepsilon)] \leq O(\varepsilon\delta),$$

If we take $r^{(L)} = O(\frac{1}{\varepsilon^2} \log \frac{1}{\delta})$,

$$\Pr[\|\sum_{u \in \mathcal{N}(v)} \frac{\partial M_{Lvu}}{\partial \boldsymbol{\theta}}(\boldsymbol{z}_v^{(L-1)}, \boldsymbol{z}_u^{(L-1)}, \boldsymbol{e}_{vu}, \boldsymbol{\theta})$$

$$- \frac{\deg(v)}{r^{(L)}} \sum_{u \in \mathcal{S}^{(L)}} \frac{\partial M_{Lvu}}{\partial \boldsymbol{\theta}}(\boldsymbol{z}_v^{(L-1)}, \boldsymbol{z}_u^{(L-1)}, \boldsymbol{e}_{vu}, \boldsymbol{\theta})\| \geq O(\varepsilon)] \leq O(\delta), \tag{24}$$

$$\Pr[\|\sum_{u \in \mathcal{N}(v)} \frac{\partial M_{Lvu}}{\partial \boldsymbol{z}_v^{(L-1)}}(\boldsymbol{z}_v^{(L-1)}, \boldsymbol{z}_u^{(L-1)}, \boldsymbol{e}_{vu}, \boldsymbol{\theta})\frac{\partial \boldsymbol{z}_v^{(L-1)}}{\partial \boldsymbol{\theta}}$$

$$- \frac{\deg(v)}{r^{(L)}} \sum_{u \in \mathcal{S}^{(L)}} \frac{\partial M_{Lvu}}{\partial \boldsymbol{z}_v^{(L-1)}}(\boldsymbol{z}_v^{(L-1)}, \boldsymbol{z}_u^{(L-1)}, \boldsymbol{e}_{vu}, \boldsymbol{\theta})\frac{\partial \boldsymbol{z}_v^{(L-1)}}{\partial \boldsymbol{\theta}}\| \geq O(\varepsilon)] \leq O(\delta), \tag{25}$$

$$\Pr[\|\sum_{u \in \mathcal{N}(v)} \frac{\partial M_{Lvu}}{\partial \boldsymbol{z}_u^{(L-1)}}(\boldsymbol{z}_v^{(L-1)}, \boldsymbol{z}_u^{(L-1)}, \boldsymbol{e}_{vu}, \boldsymbol{\theta})\frac{\partial \boldsymbol{z}_u^{(L-1)}}{\partial \boldsymbol{\theta}}$$

$$- \frac{\deg(v)}{r^{(L)}} \sum_{u \in \mathcal{S}^{(L)}} \frac{\partial M_{Lvu}}{\partial \boldsymbol{z}_u^{(L-1)}}(\boldsymbol{z}_v^{(L-1)}, \boldsymbol{z}_u^{(L-1)}, \boldsymbol{e}_{vu}, \boldsymbol{\theta})\frac{\partial \boldsymbol{z}_u^{(L-1)}}{\partial \boldsymbol{\theta}}\| \geq O(\varepsilon)] \leq O(\delta), \tag{26}$$

holds true by the Hoeffding's inequality. Therefore,

$$\Pr[\|\sum_{u \in \mathcal{N}(v)} \frac{\partial M_{Lvu}}{\partial \boldsymbol{\theta}}(\boldsymbol{z}_v^{(L-1)}, \boldsymbol{z}_u^{(L-1)}, \boldsymbol{e}_{vu}, \boldsymbol{\theta})$$

$$- \frac{\deg(v)}{r^{(L)}} \sum_{u \in \mathcal{S}^{(L)}} \frac{\partial M_{Lvu}}{\partial \boldsymbol{\theta}} (\hat{\boldsymbol{z}}_v^{(L-1)}, \hat{\boldsymbol{z}}_u^{(L-1)}, \boldsymbol{e}_{vu}, \boldsymbol{\theta}) \| \geq O(\varepsilon)] \leq O(\delta), \tag{27}$$

$$\Pr[\| \sum_{u \in \mathcal{N}(v)} \frac{\partial M_{Lvu}}{\partial \boldsymbol{z}_v^{(L-1)}} (\boldsymbol{z}_v^{(L-1)}, \boldsymbol{z}_u^{(L-1)}, \boldsymbol{e}_{vu}, \boldsymbol{\theta}) \frac{\partial \boldsymbol{z}_v^{(L-1)}}{\partial \boldsymbol{\theta}}$$
$$- \frac{\deg(v)}{r^{(L)}} \sum_{u \in \mathcal{S}^{(L)}} \frac{\partial M_{Lvu}}{\partial \hat{\boldsymbol{z}}_v^{(L-1)}} (\hat{\boldsymbol{z}}_v^{(L-1)}, \hat{\boldsymbol{z}}_u^{(L-1)}, \boldsymbol{e}_{vu}, \boldsymbol{\theta}) \frac{\partial \hat{\boldsymbol{z}}_v^{(L-1)}}{\partial \boldsymbol{\theta}} \| \geq O(\varepsilon)] \leq O(\delta), \tag{28}$$

$$\Pr[\| \sum_{u \in \mathcal{N}(v)} \frac{\partial M_{Lvu}}{\partial \boldsymbol{z}_u^{(L-1)}} (\boldsymbol{z}_v^{(L-1)}, \boldsymbol{z}_u^{(L-1)}, \boldsymbol{e}_{vu}, \boldsymbol{\theta}) \frac{\partial \boldsymbol{z}_u^{(L-1)}}{\partial \boldsymbol{\theta}}$$
$$- \frac{\deg(v)}{r^{(L)}} \sum_{u \in \mathcal{S}^{(L)}} \frac{\partial M_{Lvu}}{\partial \hat{\boldsymbol{z}}_u^{(L-1)}} (\hat{\boldsymbol{z}}_v^{(L-1)}, \hat{\boldsymbol{z}}_u^{(L-1)}, \boldsymbol{e}_{vu}, \boldsymbol{\theta}) \frac{\partial \hat{\boldsymbol{z}}_u^{(L-1)}}{\partial \boldsymbol{\theta}} \| \geq O(\varepsilon)] \leq O(\delta), \tag{29}$$

holds true by equations (20), (21), (22), (23), (24), (25), and (26).

Therefore, if we take $r^{(1)}, \ldots r^{(L)}$ sufficiently large, $\Pr[\| \frac{\partial \boldsymbol{z}_v^{(L)}}{\partial \boldsymbol{\theta}} - \frac{\partial \hat{\boldsymbol{z}}_v^{(L)}}{\partial \boldsymbol{\theta}} \|_F \geq \varepsilon] \leq \delta$ holds true by equations (20), (21), (22), (23), (27), (28), and (29). $\qquad\square$

*Proof of Theorem 5.* We prove this by performing mathematical induction on the number of layers.

**Base case:** It is shown that the statement holds true for $L = 1$.

If $DU_L$ is $K'$-Lipschitz continuous, $\varepsilon' = O(\varepsilon)$ in equation (15). Therefore, $r^{(L)} = O(\frac{1}{\varepsilon^2} \log \frac{1}{\delta})$ is sufficient.

**Inductive step:** It is shown that the statement holds true for $L = l + 1$ if it holds true for $L = l$.

If $DU_L$ and $\deg(v)DU_{Lvu}$ is $K'$-Lipschitz continuous, $\varepsilon' = O(\varepsilon)$ in equation (20) and $\varepsilon'' = O(\varepsilon)$ in equation (21). Therefore, $r^{(L)} = O(\frac{1}{\varepsilon^2} \log \frac{1}{\delta})$ and $r^{(1)}, \ldots, r^{(L-1)} = O(\frac{1}{\varepsilon^2}(\log \frac{1}{\varepsilon} + \log \frac{1}{\delta}))$ are sufficient. $\qquad\square$

*Proof of Theorem 6.* We show that one-layer GraphSAGE-GCN whose activation function is not constant cannot be approximated in constant time if $\|x_v\|_2$ or $\|\theta\|_2$ are not bounded. There exists $a \in \mathbb{R}$ such that $\sigma(a) \neq \sigma(0)$ because $\sigma$ is not constant. We consider the following two types of inputs:

- $G$ is the clique $K_n$, $\boldsymbol{W}^{(1)} = 1$, and $\boldsymbol{x}_i = 0$ for all nodes $i \in \mathcal{V}$.

- $G$ is the clique $K_n$, $\boldsymbol{W}^{(1)} = 1$, $\boldsymbol{x}_i = 0 (i \neq v)$ for some $v \in \mathcal{V}$, and $\boldsymbol{x}_v = an$.

Then, for the former input, $\boldsymbol{z}_v^{(1)} = \sigma(0)$. For the latter type of inputs, $\boldsymbol{z}_v^{(1)} = \sigma(a)$. Let $\mathcal{A}$ be an arbitrary constant algorithm and $C$ be the number of queries $\mathcal{A}$ makes when we set $\varepsilon = |\sigma(a) - \sigma(0)|/3$. When $\mathcal{A}$ calculates the embedding of $u \neq v \in \mathcal{V}$, the states of all nodes but $u$ are symmetrical until $\mathcal{A}$ makes a query about that node. Therefore, if $n$ is sufficiently large, $\mathcal{A}$ does not make any query about $v$ with high probability (i.e., at least $(1 - \frac{1}{n-1})^C$). If $\mathcal{A}$ does not make any query about $v$, the state of $\mathcal{A}$ is the same for both types of inputs. If the approximation error is less than $\varepsilon$ for the first type of inputs, the approximation error is larger than $\varepsilon$ for the second type of inputs by the triangle inequality and vice versa. Therefore, $\mathcal{A}$ fails to approximate the embeddings of either type of inputs with the absolute error of at most $\varepsilon$. As for $\boldsymbol{\theta}$, we set $\boldsymbol{W}^{(1)} = an$ and $\boldsymbol{x}_v = 1$ for the second type of inputs. Then, the same argument follows. $\qquad\square$

*Proof of Theorem 7.* We consider the one-layer GraphSAGE-GCN with ReLU and normalization (i.e., $\sigma(\boldsymbol{x}) = \text{RELU}(\boldsymbol{x})/\|\text{RELU}(\boldsymbol{x})\|_2$). We use the following two types of inputs:

- $G$ is the clique $K_n$, $\boldsymbol{W}^{(1)}$ is the identity matrix $\boldsymbol{I}_2$, $\boldsymbol{x}_i = (0,0)^\top (i \neq v)$ for some node $v \in \mathcal{V}$, and $\boldsymbol{x}_v = (1,0)^\top$.

- $G$ is the clique $K_n$, $\boldsymbol{W}^{(1)}$ is the identity matrix $\boldsymbol{I}_2$, $\boldsymbol{x}_i = (0,0)^\top (i \neq v)$ for some node $v \in \mathcal{V}$, and $\boldsymbol{x}_v = (0,1)^\top$.

Then, for the former type of inputs, $\boldsymbol{h}_i = (1/n, 0)^\top$, $\boldsymbol{z}_i = (1,0)^\top$, and $\frac{\partial z_{i2}}{\partial W_{21}} = 1$ for all $i \in \mathcal{V}$. For the latter type of inputs, $\boldsymbol{h}_i = (0, 1/n)^\top$, $\boldsymbol{z}_i = (0,1)^\top$, and $\frac{\partial z_{i2}}{\partial W_{21}} = 0$ for all $i \in \mathcal{V}$. Let $\mathcal{A}$ be an arbitrary constant algorithm and $C$ be the number of queries $\mathcal{A}$ makes when we set $\varepsilon = 1/3$. When $\mathcal{A}$ calculates the embedding or gradient of $u \neq v \in \mathcal{V}$, the states of all nodes but $u$ are symmetrical until $\mathcal{A}$ makes a query about that node. Therefore, if $n$ is sufficiently large, $\mathcal{A}$ does not make any query about $v$ with high probability (i.e., at least $(1 - \frac{1}{n-1})^C$). If $\mathcal{A}$ does not make any query about $v$, the state of $\mathcal{A}$ is the same for both types of inputs. If the approximation error is less than $\varepsilon$ for the first type of inputs, the approximation error is larger than $\varepsilon$ for the second type of inputs by the triangle inequality and vice versa. Therefore, $\mathcal{A}$ fails to approximate the embeddings and gradients of either type of inputs with the absolute error of at most $\varepsilon$. □

*Proof of Theorem 8.* We consider the one-layer GraphSAGE-GCN with ReLU (i.e., $\sigma(\boldsymbol{x}) = \text{RELU}(\boldsymbol{x})$). We use the following two types of inputs:

- $G$ is the clique $K_n$, $\boldsymbol{W}^{(1)} = (-1,1)$, $\boldsymbol{x}_i = (1,1)^\top (i \neq v)$ for some node $v \in \mathcal{V}$, and $\boldsymbol{x}_v = (1,2)^\top$.

- $G$ is the clique $K_n$, $\boldsymbol{W}^{(1)} = (-1,1)$, $\boldsymbol{x}_i = (1,1)^\top (i \neq v)$ for some node $v \in \mathcal{V}$, and $\boldsymbol{x}_v = (1,0)^\top$.

Then, for the former type of inputs, $\text{MEAN}(\{\boldsymbol{x}_u \mid u \in \mathcal{N}(v)\}) = (1, 1 + \frac{1}{n})^\top$, $\boldsymbol{h}_v = \boldsymbol{z}_v = \frac{1}{n}$, and $\frac{\partial \boldsymbol{z}_v}{\partial \boldsymbol{W}} = (1, 1 + \frac{1}{n})$ for all $i \in \mathcal{V}$. For the latter type of inputs, $\text{MEAN}(\{\boldsymbol{x}_u \mid u \in \mathcal{N}(v)\}) = (1, 1 - \frac{1}{n})^\top$, $\boldsymbol{h}_v = -\frac{1}{n}$, $\boldsymbol{z}_v = 0$, and $\frac{\partial \boldsymbol{z}_v}{\partial \boldsymbol{W}} = (0,0)$ for all $i \in \mathcal{V}$. Let $\mathcal{A}$ be an arbitrary constant algorithm and $C$ be the number of queries $\mathcal{A}$ makes when we set $\varepsilon = 1/3$. When $\mathcal{A}$ calculates the gradient of $u \neq v \in \mathcal{V}$, the states of all nodes but $u$ are symmetrical until $\mathcal{A}$ makes a query about that node. Therefore, if $n$ is sufficiently large, $\mathcal{A}$ does not make any query about $v$ with high probability (i.e., at least $(1 - \frac{1}{n-1})^C$). If $\mathcal{A}$ does not make any query about $v$, the state of $\mathcal{A}$ is the same for both types of inputs. If the approximation error is less than $\varepsilon$ for the first type of inputs, the approximation error is larger than $\varepsilon$ for the second type of inputs by the triangle inequality and vice versa. Therefore, $\mathcal{A}$ fails to approximate the gradients of either type of inputs with the absolute error of at most $\varepsilon$. □

*Proof of Theorem 9.* We consider the one-layer GraphSAGE-pool whose activation function satisfies $\sigma(1) \neq \sigma(0)$ and the following two types of inputs:

- $G$ is the clique $K_n$, $\boldsymbol{W}^{(1)} = 1$, $\boldsymbol{b} = 0$, and $\boldsymbol{x}_i = 0$ for all nodes $v \in \mathcal{V}$.

- $G$ is the clique $K_n$, $\boldsymbol{W}^{(1)} = 1$, $\boldsymbol{b} = 0$, $\boldsymbol{x}_i = 0 (i \neq v)$ for some node $v \in \mathcal{V}$, and $\boldsymbol{x}_v = 1$.

Then, for the former type of inputs, $\boldsymbol{z}_i = \sigma(0)$ for all $i \in \mathcal{V}$. For the latter type of inputs, $\boldsymbol{z}_i = \sigma(1)$ for all $i \in \mathcal{V}$. Let $\mathcal{A}$ be an arbitrary constant algorithm and $C$ be the number of queries $\mathcal{A}$ makes when we set $\varepsilon = |\sigma(1) - \sigma(0)|/3$. When $\mathcal{A}$ calculates the embedding of $u \neq v \in \mathcal{V}$, the states of all nodes but $u$ are symmetrical until $\mathcal{A}$ makes a query about that node. Therefore, if $n$ is sufficiently large, $\mathcal{A}$ does not make any query about $v$ with high probability (i.e., at least $(1 - \frac{1}{n-1})^C$). If $\mathcal{A}$ does not make any query about $v$, the state of $\mathcal{A}$ is the same for both types of inputs. If the approximation error is less than $\varepsilon$ for the first type of inputs, the approximation error is larger than $\varepsilon$ for the second type of inputs by the triangle inequality and vice versa. Therefore, $\mathcal{A}$ fails to approximate the embeddings of either type of inputs with the absolute error of at most $\varepsilon$. □

*Proof of Lemma 10.* We consider the one-layer GCN whose activation function satisfies $\sigma(1) \neq \sigma(0)$. We use the following two types of inputs:

- $G$ is a star graph, where $v \in \mathcal{V}$ is the center of $G$, $\boldsymbol{W}^{(1)} = 1$, and all features are 0.

- $G$ is a star graph, where $v \in \mathcal{V}$ is the center of $G$, $\boldsymbol{W}^{(1)} = 1$, and the features of $\sqrt{2n}$ leafs are 1 and the features of other nodes are 0.

Then, for the former type of inputs, $\boldsymbol{z}_v = \sigma(0)$. For the latter type of inputs, $\boldsymbol{z}_v = \sigma(1)$. Let $\mathcal{A}$ be an arbitrary constant algorithm and $C$ be the number of queries $\mathcal{A}$ makes when we set $\varepsilon = |\sigma(1) - \sigma(0)|/3$. When $\mathcal{A}$ calculates the embedding of $u \in \mathcal{V}$ that $\boldsymbol{x}_u = 0$, the states of all nodes but $u$ are symmetrical until $\mathcal{A}$ makes a query about that node. Therefore, if $n$ is sufficiently large, $\mathcal{A}$ does not make any query about $v$ with high probability (i.e., at least $(1 - \frac{\sqrt{2n}}{n-1})^C$). If $\mathcal{A}$ does not make any query about $v$, the state of $\mathcal{A}$ is the same for both types of inputs. If the approximation error is less than $\varepsilon$ for the first type of inputs, the approximation error is larger than $\varepsilon$ for the second type of inputs by the triangle inequality and vice versa. Therefore, $\mathcal{A}$ fails to approximate the embeddings of either type of inputs with the absolute error of at most $\varepsilon$. $\qquad\square$

We prove the following lemma to prove Proposition 11.

**Lemma 17.** *If Assumptions 1 holds true and $\sigma$ is Lipschitz continuous, $\|\boldsymbol{z}_v^{(l)}\|_2$ and $\|\boldsymbol{z}_v^{(l)}\|_2$ ($l = 1, \ldots, L$) of the GAT model are bounded by a constant*

*Proof of Lemma 17.* We prove this by performing mathematical induction on the number of layers. The norm of the input of the first layer is bounded by Assumption 1. If $\|\boldsymbol{z}_u^{(l-1)}\|_2$ and $\|\hat{\boldsymbol{z}}_u^{(l-1)}\|_2$ are bounded for all $u \in \mathcal{V}$, $\|\boldsymbol{h}_v^{(l)}\|_2$ and $\|\hat{\boldsymbol{h}}_v^{(l)}\|_2$ are bounded because $\boldsymbol{h}_v^{(l)}$ and $\hat{\boldsymbol{h}}_v^{(l)}$ are the weighted sum of $\boldsymbol{z}_u^{(l-1)}$ and $\hat{\boldsymbol{z}}_u^{(l-1)}$. Therefore, $\|\boldsymbol{z}_u^{(l)}\|_2$ and $\|\hat{\boldsymbol{z}}_u^{(l)}\|_2$ are bounded because $U_l$ is continuous. $\qquad\square$

*Proof of Proposition 11.* We prove the theorem by performing mathematical induction on the number of layers $L$.

**Base case:** It is shown that the statement holds true for $L = 1$.

Because $U_L$ is Lipschitz continuous,
$$\forall \boldsymbol{z}_v, \boldsymbol{h}_v, \boldsymbol{h}_v', \boldsymbol{\theta}, \|\boldsymbol{h}_v - \boldsymbol{h}_v'\|_2 < O(\varepsilon) \Rightarrow \|U_L(\boldsymbol{z}_v, \boldsymbol{h}_v, \boldsymbol{\theta}) - U_L(\boldsymbol{z}_v, \boldsymbol{h}_v', \boldsymbol{\theta})\|_2 < \varepsilon. \tag{30}$$
Let $e_u = \exp(\text{LEAKYRELU}(\boldsymbol{a}^{(l)\top}[\boldsymbol{W}^{(0)}\boldsymbol{z}_v^{(0)}, \boldsymbol{W}^{(0)}\boldsymbol{z}_u^{(0)}u]))$. Then,
$$\hat{\boldsymbol{h}}_v^{(L)} - \boldsymbol{h}_v^{(L)}$$
$$= \sum_{u \in \mathcal{S}^{(L)}} \hat{\alpha}_{vu}\boldsymbol{z}_u^{(0)} - \sum_{u \in \mathcal{N}(v)} \alpha_{vu}\boldsymbol{z}_u^{(0)}$$
$$= \frac{1}{r^{(L)}} \sum_{u \in \mathcal{S}^{(L)}} \frac{e_u}{\frac{1}{r^{(L)}}\sum_{u' \in \mathcal{S}^{(L)}} e_{u'}} \boldsymbol{z}_u^{(0)} - \frac{1}{\deg(v)} \sum_{u \in \mathcal{N}(v)} \frac{e_u}{\frac{1}{\deg(v)}\sum_{u' \in \mathcal{N}(v)} e_{u'}} \boldsymbol{z}_u^{(0)}.$$
Let $x_k$ be the $k$-th sample in $\mathcal{S}^{(L)}$ and $X_k = e_{x_k}$. Then,
$$\mathbb{E}[X_k] = \frac{1}{\mathcal{N}(v)} \sum_{u \in \mathcal{N}(v)} e_u. \tag{31}$$

There exists a constant $c > 0, C > 0$ such that for any input satisfying Assumption 1,
$$c < |X_k| < C, \tag{32}$$
because $\|\boldsymbol{z}_v^{(0)}\|_2, \|\boldsymbol{z}_{x_k}^{(0)}\|_2, \|\boldsymbol{W}^{(0)}\|_F$, and $\|\boldsymbol{a}^{(0)}\|_2$ are bounded by Assumption 1. Therefore, if we take $r^{(L)} = O(\frac{1}{\varepsilon^2}\log\frac{1}{\delta})$ samples,
$$\Pr[|\frac{1}{r^{(L)}} \sum_{u \in \mathcal{S}^{(L)}} e_u - \frac{1}{\mathcal{N}(v)} \sum_{u \in \mathcal{N}(v)} e_u| \geq O(\varepsilon)] \leq O(\delta).$$

by the Hoeffding's inequality and equations (31) and (32). Because $f(x) = 1/x$ is Lipschitz continuous in $x > c > 0$,

$$\Pr[|\frac{1}{\frac{1}{r^{(L)}}\sum_{u\in\mathcal{S}^{(L)}}e_u} - \frac{1}{\frac{1}{\mathcal{N}(v)}\sum_{u\in\mathcal{N}(v)}e_u}| \geq O(\varepsilon)] \leq O(\delta) \tag{33}$$

Let

$$Y_k = \frac{e_{x_k}}{\frac{1}{\mathcal{N}(v)}\sum_{u'\in\mathcal{N}(v)}e_{u'}}\mathbf{z}_{x_k}^{(0)}.$$

Then,

$$\mathbb{E}[Y_k] = \frac{1}{\mathcal{N}(v)}\sum_{u\in\mathcal{N}(v)}\frac{e_u}{\frac{1}{\mathcal{N}(v)}\sum_{u'\in\mathcal{N}(v)}e_{u'}}\mathbf{z}_u^{(0)}. \tag{34}$$

There exists a constant $C' \in \mathbb{R}$ such that for any input satisfying Assumption 1,

$$\|Y_k\|_2 < C' \tag{35}$$

holds true because $\|\mathbf{z}_u^{(0)}\|_2$ are bounded, and $c < |e_u| < C$. Therefore, if we take $r^{(L)} = O(\frac{1}{\varepsilon^2}\log\frac{1}{\delta})$ samples,

$$\Pr[\|\frac{1}{r^{(L)}}\sum_{u\in\mathcal{S}^{(L)}}\frac{e_u}{\frac{1}{\deg(v)}\sum_{u'\in\mathcal{N}(v)}e_{u'}}\mathbf{z}_u^{(0)}$$
$$-\frac{1}{\deg(v)}\sum_{u\in\mathcal{N}(v)}\frac{e_u}{\frac{1}{\deg(v)}\sum_{u'\in\mathcal{N}(v)}e_{u'}}\mathbf{z}_u^{(0)}\|_2 \geq O(\varepsilon)] \leq O(\delta) \tag{36}$$

holds true by the Hoeffding's inequality and equations (34) and (35). Therefore,

$$\Pr[\|\frac{1}{r^{(L)}}\sum_{u\in\mathcal{S}^{(L)}}\frac{e_u}{\frac{1}{r^{(L)}}\sum_{u'\in\mathcal{S}^{(L)}}e_{u'}}\mathbf{z}_u^{(0)}$$
$$-\frac{1}{\deg(v)}\sum_{u\in\mathcal{N}(v)}\frac{e_u}{\frac{1}{\deg(v)}\sum_{u'\in\mathcal{N}(v)}e_{u'}}\mathbf{z}_u^{(0)}\|_2 \geq O(\varepsilon)] \leq O(\delta) \tag{37}$$

holds true by the triangle inequality and equations (33) and (36), and $\Pr[\|\hat{\mathbf{z}}_v^{(L)} - \mathbf{z}_v^{(L)}\|_2 \geq \varepsilon] \leq \delta$ holds true by equations (30) and (37).

**Inductive step:** It is shown that the statement holds true for $L = l+1$ if it holds true for $L = l$.

Because $U_L$ is Lipschitz continuous,

$$\forall \mathbf{z}_v, \mathbf{h}_v, \mathbf{h}'_v, \boldsymbol{\theta}, \|\mathbf{h}_v - \mathbf{h}'_v\|_2 < O(\varepsilon) \Rightarrow \|U_L(\mathbf{z}_v, \mathbf{h}_v, \boldsymbol{\theta}) - U_L(\mathbf{z}_v, \mathbf{h}'_v, \boldsymbol{\theta})\|_2 < \varepsilon \tag{38}$$

holds true.

$$\Pr[\|\frac{1}{r^{(L)}}\sum_{u\in\mathcal{S}^{(L)}}\frac{e_u}{\frac{1}{r^{(L)}}\sum_{u'\in\mathcal{S}^{(L)}}e_{u'}}\mathbf{z}_u^{(L-1)}$$
$$-\frac{1}{\deg(v)}\sum_{u\in\mathcal{N}(v)}\frac{e_u}{\frac{1}{\deg(v)}\sum_{u'\in\mathcal{N}(v)}e_{u'}}\mathbf{z}_u^{(L-1)}\|_2 \geq O(\varepsilon)] \leq O(\delta) \tag{39}$$

holds true by the same argument as the base step. If we take $r^{(1)}, \ldots, r^{(L-1)} = O(\frac{1}{\varepsilon^2}(\log\frac{1}{\varepsilon} + \log\frac{1}{\delta}))$ samples,

$$\Pr[\|\hat{\mathbf{z}}_u^{(L-1)} - \mathbf{z}_u^{(L-1)}\|_2 \geq O(\varepsilon)] \leq O(\varepsilon\delta) \tag{40}$$

holds true by the induction hypothesis. Therefore, $\Pr[\|\hat{\mathbf{z}}_v^{(L)} - \mathbf{z}_v^{(L)}\|_2 \geq \varepsilon] \leq \delta$ holds true by equations (38), (39), and (40).

$\square$

## D    COMPUTATIONAL MODEL ASSUMPTIONS

In this study, we model our algorithm as an oracle machine that can make queries about the input and measure the complexity by query complexity. Modeling our algorithm as an oracle machine and measuring the complexity by query complexity are reasonable owing to the following reasons:

- In a realistic setting, data is stored in a storage or cloud and we may not be able to load all the information of a huge network on to the main memory. Sometimes the network is constructed for access to the information on demand (e.g., in web graph mining, the edge information is retrieved when queried). In such cases, reducing the number of queries is crucial because accessing storage or cloud is very expensive.

- Our algorithm executes a constant number of elementary operations of $O(\log n)$ bits (e.g., accessing the $O(n)$-th address, sampling one element from $O(n)$ elements). Therefore, if we assume that these operations can be done in constant time, the total computational complexity of our algorithms will be constant. This assumption is natural because most computers in the real-world can handle $64$ bit integers at once and most of the network data contain less than $2^{64} \approx 10^{19}$ nodes.

- Even if the above assumption is not satisfied, our algorithm can be executed in $O(\log n)$ time in terms of the strict meaning of computational complexity. This indicates that our algorithm is still sub-linear and therefore it scales well. It should be noted that it is impossible to access even a single node in $o(\log n)$ time in the strict meaning of computational complexity because we cannot distinguish $n$ nodes with $o(\log n)$ bits. Therefore, our algorithm has optimal complexity with respect to the number of nodes $n$.

## E    GRAPH EMBEDDING

In this study, we model our algorithm as an oracle machine that can make queries about the input and measure the complexity by query complexity. Modeling our algorithm as an oracle machine and measuring the complexity by query complexity are reasonable owing to the following reasons:

- In a realistic setting, data is stored in a storage or cloud, and we may not be able to load all the information of a huge network on to the main memory. Sometimes the network is constructed for access to the information on demand (e.g., in web graph mining, the edge information is retrieved when queried). In such cases, reducing the number of queries is crucial because accessing storage or cloud is very expensive.

- Our algorithm executes a constant number of elementary operations of $O(\log n)$ bits (e.g., accessing the $O(n)$-th address, sampling one element from $O(n)$ elements). Therefore, if we assume that these operations can be done in constant time, the total computational complexity of our algorithms will be constant. This assumption is natural because most computers in the real-world can handle $64$ bit integers at once, and most of the network data contain less than $2^{64} \approx 10^{19}$ nodes.

- Even if the above assumption is not satisfied, our algorithm can be executed in $O(\log n)$ time in terms of the strict meaning of computational complexity. This indicates that our algorithm is still sub-linear and therefore, it scales well. It should be noted that it is impossible to access even a single node in $o(\log n)$ time in the strict meaning of computational complexity because we cannot distinguish $n$ nodes with $o(\log n)$ bits. Therefore, our algorithm has optimal complexity with respect to the number of nodes $n$.

Table 2: Time complexity of embedding algorithms. $\Delta$ denotes the maximum degree. It should be noted that that in the dense graph, $O(m) = O(n^2)$ by definition.

|  | Sparse | Dense |
|---|---|---|
| Proposed | $O(\frac{1}{\varepsilon^{2L}}(\log \frac{1}{\varepsilon} + \log \frac{1}{\delta})^{L-1} \log \frac{1}{\delta})$ | $O(\frac{1}{\varepsilon^{2L}}(\log \frac{1}{\varepsilon} + \log \frac{1}{\delta})^{L-1} \log \frac{1}{\delta})$ |
| Exact | $O(\Delta^L)$ | $O(mL) = O(n^2 L)$ |

## F  TIME COMPLEXITY

We summarize the time complexity of approximation and exact algorithms in Table 2. Let $\mathcal{B}_L = \{v\}$ and $\mathcal{B}_l = \bigcup_{u \in \mathcal{B}_{l+1}} \mathcal{N}(u)$ $(l = 0, \dots, L-1)$. In other words, $\mathcal{B}_{L-k}$ is the $k$-hop neighbors of node $v$. We need all features of $\mathcal{B}_0$ to calculate the exact embedding of node $v$. If the graph is sparse, the size of $\mathcal{B}_{L-k}$ grows exponentially with respect to $k$ because $\deg(u)$ nodes are added to $\mathcal{B}_{L-k+1}$ for each node $u \in \mathcal{B}_{L-k}$. Namely, it is bounded by $\Delta^k$. If the graph is dense, $\mathcal{B}_{L-k} \approx \mathcal{V}$ $(1 \leq k \leq L)$ Therefore, complexity is linear with respect to the number of layers $L$ and edges $m$. In contrast, the approximation algorithm runs in constant time irrespective of the density of the input graph.

