# OpenReview forum: "Constant Time Graph Neural Networks"
_ICLR.cc/2020/Conference — Reject_

### Official Review · AnonReviewer3 · 2019-10-14
**Official Blind Review #3**

**Rating:** 3

**Review:**

In this paper, the authors propose a constant-time approximation for graph convolution operation via theoretical analysis on the number of sampling from each neighbor. The authors prove that both node embedding and gradient can be approximated via constant number of samples among the neighbors. Extensive experiments are carried out to verify the correctness of the proposed bounds.

Strength:
1. The authors establish rigorous bounds on the number of required neighbors to sample to guarantee good approximation of the graph convolution operations. The bounds are established on both nodes embedding and gradients.
2. The authors corroborate the theoretical results with extensive experiments. The experiments are carried under the case both when the assumption holds and not.

Weakness:
1. Though the bounds do not depend on number of nodes, it does have an exponential dependence on the number of layers. For common GCN with L=2, even \epsilon = 0.1 requires more than 10^4 samples. As a result, the bound is less practical in providing guidance for real-world application. Moreover, the bound does not depend on the embedding dimension at all. Is this to be expected?
2. The authors only provide constant bound for convolution on individual nodes. For the final metrics like classification accuracy, we need to guarantee good approximation for all node’s convolution. In this case, the bounds will definitely depend on number of nodes at least through the union bounds among all nodes.
3. The proposed method is exactly the uniform neighbor sampling methods. There is little contribution on the methodology side from the paper. Also, it would be interesting if the authors could provide theoretical analysis for adaptive sampling methods as well.
4. The authors provide in-approximability results under certain conditionals. However, uniform sampling methods works well empirically under this case as well. It would be better if the authors could explain the gap between theory and empirical performance.

Detailed Comments:
1. Algorithm 2, please define notation like O(v<-v).


**Experience Assessment:**

I have published one or two papers in this area.

**Review Assessment: Checking Correctness Of Derivations And Theory:**

I assessed the sensibility of the derivations and theory.

**Review Assessment: Checking Correctness Of Experiments:**

I assessed the sensibility of the experiments.

**Review Assessment: Thoroughness In Paper Reading:**

I read the paper at least twice and used my best judgement in assessing the paper.

---

> ### Author Response · Authors · 2019-11-11
> **Author Response to Official Blind Review #3**
>
> Thank you for your constructive comments.
>
> > 1. Though the bounds do not depend on number of nodes, it does have an exponential dependence on the number of layers. For common GCN with L=2, even \epsilon = 0.1 requires more than 10^4 samples. As a result, the bound is less practical in providing guidance for real-world application.
>
> Our analysis is sometimes looser than the errors in empirical cases because we consider the worst case with mild assumptions.
>
> This is common with PAC learning literature and many other constant time algorithms. For example, the query complexities of Nguyen and Onak's constant time algorithms for the maximum matching problem and the minimum dominating set problem depend on $O((1/\varepsilon)^{2^k})$ ($\varepsilon$ is the approximation error, and $k$ is the maximum degree of input graphs) [Nguyen and Onak, FOCS 2008], which is extremely large in practice, whereas the approximation algorithms work far more efficiently than this bound in practice. However, their theoretical bounds are nonetheless informative because the assumptions are general, and these bounds are constants with respect to the number of nodes and edges. The usefulness of their theoretical results has been proved by many other works following this work.
>
> The tightness of theoretical error bound and generality is in a trade-off relation. If one wants to obtain tighter bound, one should make more assumptions that are characteristic of the problem. For example, if one assumes that the model architecture is GraphSAGE-mean, the input graph is sampled from the Erdős-Rényi model, and the node features are sampled from i.i.d. Gaussian distribution, in addition to Assumptions 1 to 5, then the error bound does not depend on the number of layers by trivial discussion because what we only have to estimate is the parameters of the Gaussian distribution. (They are strong assumptions, and the empirical situations are usually placed between it and our assumptions.)
>
> We analyze general cases by imposing mild assumptions in this paper because this is the first work discussing the theoretical approximation errors of node sampling. As AnonReviewer1 pointed out, our analysis provides an interesting understanding of an approach that is commonly used in practice, and we consider our work is useful for further analysis for more specific situations to obtain tighter bounds.
>
> > the bound does not depend on the embedding dimension
>
> In this paper, we fix the architecture of the model structure, as we mention in the problem formulation section, and we treat the dimensions of intermediate embeddings and final embeddings as a constant. As our proofs in the appendix shows, the error bounds depend quasi linearly on the final embedding dimension.
>
>
> > 2. The authors only provide constant bound for convolution on individual nodes. For the final metrics like classification accuracy, we need to guarantee good approximation for all node’s convolution. In this case, the bounds will definitely depend on number of nodes at least through the union bounds among all nodes.
>
> Exactly. It should be noted that the inference for a single node is also useful in the online setting, as we noted in the footnote of the introduction because the delay of the response of the inference for a single node matters in this case.
>
> If one needs to obtain a good approximation for all nodes, the computational complexity depends on the number of nodes. This is necessary because the output size itself depends on the number of nodes. Note that even in this case, the dependency of our analysis to the input size is optimal. Especially, the complexity does not depend on the number of edges, whereas exact computation requires at least |E| operations, which is computationally heavy if the input graph is dense.

---

> ### Author Response · Authors · 2019-11-11
> **Author Response to Official Blind Review #3**
>
> > 3. The proposed method is exactly the uniform neighbor sampling methods. There is little contribution on the methodology side from the paper. Also, it would be interesting if the authors could provide theoretical analysis for adaptive sampling methods as well.
>
> Our problem setting is general, and we do not make any assumptions about the input graph. Therefore, in this setting, non-uniform sampling does not work because we allow adversarial inputs. This is supported by Theorem 3, which shows that the uniform sampling scheme is optimal in terms of query complexity. This is common with Hayashi and Yoshida’s constant time algorithms for minimizing quadratic functions [Hayashi and Yoshida, NIPS 2016] and fitting low-rank tensors [Hayashi and Yoshida, NIPS 2017], where they succeeded in providing constant time algorithms using uniform sampling. If we make task-specific assumptions about the input graph/features (e.g., the input graph is sampled from the stochastic block model, and the features are sampled from i.i.d. distribution in each cluster), adaptive sampling techniques may work.
>
> In this paper, we do not consider such task-specific assumptions but use a general and widely-applicable setting because the task-specific assumptions are in general strong and narrow the score of application. This is the first work to analyze the theoretical approximation errors of node sampling, and many other task-specific analysis can stem from this work. Note that if we sample nodes without replacement, the approximation error may decrease in practice, but the theoretical rate does not change.
>
>
> > 4. The authors provide in-approximability results under certain conditionals. However, uniform sampling methods works well empirically under this case as well. It would be better if the authors could explain the gap between theory and empirical performance.
>
> The reasons why uniform sampling also succeeds in inapproximable settings are two-fold. The first reason is that the input graphs/features have nice properties in some cases (e.g., the input graph is sampled from the stochastic block model.) The second reason is that one uses more samples as the size of input graphs increases. We prove that some models cannot be approximated by a constant number of samples, but such networks may be approximated by, say, linear number of samples with respect to the input size. Our experiments support this fact. The number of samples is usually determined using validation data (e.g., cross-validation) in practice. So one may use more samples implicitly when the input data are large and dense. Our theory provides theoretical justification for determining the number of samples beforehand for various situations.
>
>
> > 1. Algorithm 2, please define notation like O(v<-v).
>
> $\mathcal{O}(v \leftarrow u)$ means calling the oracle $\mathcal{O}$ with the same parameters as the current function except for $v$, which is replaced by $u$. We added the notations for Algorithm 2. (see the fifth line of Section 4.1) Thank you for your suggestion.

---

> ### Comment · AnonReviewer3 · 2019-11-12
> **Thanks for the response**
>
> Thank you for the response. However the response addresses some of my concern. The issue on the looseness of bound makes it provide little guidance for any real applications. I will remain my initial evaluation.

---

### Official Review · AnonReviewer1 · 2019-10-22
**Official Blind Review #1**

**Rating:** 6

**Review:**

In this paper, the authors provide a theoretical framework for characterizing the approximation guarantees provided by node sampling to estimate embeddings in various GNN architectures. In particular, they prove several PAC learning-style bounds on the embedding and gradient estimation when using node sampling approaches. They also observe that since the number of nodes selected for sampling is not dependent on the size of the graph, this amounts to a constant time operation for determining embedding and gradient estimates.

Importantly, a variety of existing works have already proposed node sampling (“constant-time GNNs”), so the contribution of this work lies in the theoretical bounds. A moderate set of experiments suggests the empirical behavior follows the theoretical expectations.

Overall (and described in the comments below), I believe this paper provides some interesting theoretical results for an approach that is widely-used in practice. However, I believe the title and abstract are misleading; this paper does not suggest some new sampling strategy; it only gives a theoretical approximation error for existing approaches and network architectures.

Comments

While the authors do make this clear later in the paper, the contributions of this work are only the theoretical bounds; the node sampling algorithms have been proposed in existing work. Both the title and abstract are misleading.

Still, the theoretical bounds do provide an interesting understanding of an approach which is commonly used in practice.

It would be helpful to give the intuition behind the assumptions and proofs. Currently, it is difficult for a non-expert to follow the reasoning and practical implications of much of this work.

Currently, it is not clear if these results are “constructive”, in the sense that they do not obviously suggest some new sampling strategy, etc. Of course, it is still interesting to characterize commonly-used practices, but one does wish the theory suggested, say, a better sampling strategy, etc.

Typos, etc.

“r”, the number of sampled nodes, is not defined in the main body of the paper.

The text in Footnote 1 is a little confusing, in that the text refers to approximating the embedding, while the text in the footnote describes a prediction problem. A reader could easily interpret this to mean that the error on the downstream prediction problem will be bounded (which is not what this work shows).

The text mentions that O_G (v,i) refers to the i^th “neighborhood” of v. Presumably, this should be “neighbor”.

“we take” -> “We take”


**Experience Assessment:**

I have published one or two papers in this area.

**Review Assessment: Checking Correctness Of Derivations And Theory:**

I assessed the sensibility of the derivations and theory.

**Review Assessment: Checking Correctness Of Experiments:**

I carefully checked the experiments.

**Review Assessment: Thoroughness In Paper Reading:**

I read the paper at least twice and used my best judgement in assessing the paper.

---

> ### Author Response · Authors · 2019-11-11
> **Author Response to Official Blind Review #1**
>
> Thank you for your constructive comments.
>
> We changed the title to “On the approximation errors of node sampling for graph neural networks” and abstract following your suggestion. Since we could not change the title on OpenReview at this moment, we will ask the program chairs to change its title after acceptance.
>
> > better sampling strategy, etc.
> Our problem setting is general, and we do not make any assumptions about the input graph. Therefore, in this setting, non-uniform sampling does not work because we allow adversarial inputs. This is supported by Theorem 3, which shows that the uniform sampling scheme is optimal in terms of query complexity. This is common with Hayashi and Yoshida’s constant time algorithms for minimizing quadratic functions [Hayashi and Yoshida, NIPS 2016] and fitting low-rank tensors [Hayashi and Yoshida, NIPS 2017], where they succeeded in providing constant time algorithms using uniform sampling. If we make task-specific assumptions about the input graph/features (e.g., the input graph is sampled from the stochastic block model, and the features are sampled from i.i.d. distribution in each cluster), adaptive sampling techniques may work.
>
> In this paper, we do not consider such task-specific assumptions but use a general and widely-applicable setting because the task-specific assumptions are in general strong and narrow the scope of application. This is the first work to analyze the theoretical approximation errors of node sampling, and many other task-specific analysis can stem from this work. Note that if we sample nodes without replacement, the approximation error may decrease in practice, but the theoretical rate does not change.
>
>
> > Practical implication of the analysis
> The number of samples is usually determined using validation data (e.g., cross-validation) in practice. In contrast, our theory (including our theoretical bounds and inapproximability results) provides theoretical justification for the number of samples needed and helps determine the number of samples without cross-validation for various situations.
>
> > “r”, the number of sampled nodes, is not defined in the main body of the paper.
>
> We added the description in the revised paper. (see the third line of Section 4.1)
>
> > The text in Footnote 1 is a little confusing, in that the text refers to approximating the embedding, while the text in the footnote describes a prediction problem. A reader could easily interpret this to mean that the error on the downstream prediction problem will be bounded (which is not what this work shows).
>
> We modified the description following your suggestion.
>
> Note that our analysis can be applied to the prediction setting by considering the prediction problem as 1-dimensional embedding in [0, 1]. In that case, the prediction with the approximation is correct if the exact computation predicts correctly with margin $\varepsilon$.
>
> > The text mentions that O_G (v,i) refers to the i^th “neighborhood” of v. Presumably, this should be “neighbor”.
> > “we take” -> “We take”
>
> We fixed typos. Thank you for pointing out them.

---

> > ### Comment · AnonReviewer1 · 2019-11-15
> > **RE: Author response**
> >
> > I have read the author response and other review. My view of the paper ("weak accept") is a little higher due to an observation in the response.
> >
> > > our analysis can be applied to the prediction setting by considering the prediction problem as 1-dimensional embedding in [0, 1]. In that case, the prediction with the approximation is correct if the exact computation predicts correctly with margin \epsilon.
> >
> > After the authors point it out, this is obvious, but it didn't occur to me the first time through. I think it would be interesting to compare this prediction bound in the GCN setting with standard supervised PAC-learning bounds. It would be really nice to show that using the graph (compared to just a data table of some sort) improves the theoretical approximation guarantees.

---

### Official Review · AnonReviewer2 · 2019-11-18
**Official Blind Review #2**

**Rating:** 6

**Review:**

Paper strengths:

1. Overall it is a solid and well-written article.
2. The interesting and novel part is the theoretical demonstration that a fixed number of neighbors can approximate well some algorithms in the literature (it approximates well w.r.t to the forward embedding produced by those algorithms and to the backward gradients in relation to the parameters),
3.It also demonstrates that some algorithms are impossible to approximate in constant time.

Weaknesses:

1. They do not introduce a new method, the algorithm is message-passing in which not all the neighbors are considered - authors choose a fixed number of neighbors by sampling. This gives them constant time for the calculation of the embedding w.r.t to the number of neighbors

Main points:

Overall, the article is well supported theoretically and quite complete. They generally show that if the assumptions made are contradicted, the approximation becomes impossible.

The experiments seem quite well chosen for the cases studied. In general, the experiments in which authors want to show the impossibility of an approximation, are done exactly on the corner cases chosen for the theoretical demonstrations. However,  they also have experiments on real datasets, used in the literature.

Other observations:

It is not very clear what do they mean by "standardization layer from GCN" which intervenes in several theorems.

It may have been good to show how accuracy varies relative to the chosen error. It is not very intuitive what an embedding approximation error of e = 0.01 means. For example, to have 0.1 error and 95% confidence I would have to choose 300 neighbors.... That seems to be a very large number, how practical is that?


**Experience Assessment:**

I have published one or two papers in this area.

**Review Assessment: Checking Correctness Of Derivations And Theory:**

I assessed the sensibility of the derivations and theory.

**Review Assessment: Checking Correctness Of Experiments:**

I assessed the sensibility of the experiments.

**Review Assessment: Thoroughness In Paper Reading:**

I read the paper at least twice and used my best judgement in assessing the paper.

---

### Decision · Program_Chairs · 2019-12-19

**Decision:**

Reject

**Comment:**

There was some interest in the ideas presented, but this paper was on the borderline and ultimately not able to be accepted for publication at ICLR.

The primary reviewer concern was about the level of novelty and significance of the contribution. This was not sufficiently demonstrated.